# A unicellular cyanobacterium relies on sodium energetics to fix N$_2$

Si Tang[1], Xueyu Cheng[1], Yaqing Liu [1], Lu Liu[1], Dai Liu[1], Qi Yan[1], Jianming Zhu[1], Jin Zhou [1], Yuyang Jiang [2,3], Katrin Hammerschmidt [4] ✉ & Zhonghua Cai [1,2,5] ✉

Diazotrophic cyanobacteria can fix nitrogen gas (N$_2$) but are usually scarce in nitrogen-limited coastal waters, which poses an apparent ecological paradox. One hypothesis is that high salinities (> 10 g/L NaCl) may inhibit cyanobacterial N$_2$ fixation. However, here we show that N$_2$ fixation in a unicellular coastal cyanobacterium exclusively depends on sodium ions and is inhibited at low NaCl concentrations (< 4 g/L). In the absence of Na$^+$, cells of *Cyanothece* sp. ATCC 51142 (recently reclassified as *Crocosphaera subtropica*) upregulate the expression of *nifHDK* genes and synthesise a higher amount of nitrogenase, but do not fix N$_2$ and do not grow. We find that the loss of N$_2$-fixing ability in the absence of Na$^+$ is due to insufficient ATP supply. Additional experiments suggest that N$_2$ fixation in this organism is driven by sodium energetics and mixed-acid fermentation, rather than proton energetics and aerobic respiration, even though cells were cultured aerobically. Further work is needed to clarify the underlying mechanisms and whether our findings are relevant to other coastal cyanobacteria.

Organisms require combined nitrogen (N) for essential processes such as nucleic acid and protein synthesis. Despite the abundance of atmospheric nitrogen gas (N$_2$), bioavailable N is often scarce in nature[1]. In aquatic ecosystems, cyanobacteria can partially meet the need for N by fixing N$_2$, which impacts food chains, greenhouse gas sequestration, global climate change, and biogeochemical cycles[1,2]. This ability provides diazotrophic cyanobacteria with a competitive advantage over non-diazotrophic cyanobacteria and eukaryotic phytoplankton when environmental N is low. It enables them to exploit and dominate N-scarce environments worldwide, and can even lead to the development of dense cyanobacterial blooms when other nutrients are abundant[3]. Interestingly, in contrast to the frequently reported blooms of cyanobacterial N$_2$-fixers (note that non-N$_2$-fixers also bloom) in freshwater ecosystems, few blooms have been reported in N-limited saline coastal waters, despite their clear ecological advantage[4,5]. The

usual scarcity and low abundance of coastal N$_2$-fixing cyanobacteria in an N-deficient environment is striking and represents an ecological paradox.

One commonly proposed explanation is that high salinity levels (> 10 g/L NaCl), a fundamental difference between coastal water and freshwater ecosystems, severely inhibit N$_2$ fixation and population development of resident N$_2$-fixers[4–6]. However, most of these studies refer to multicellular N$_2$-fixing cyanobacteria and therefore much remains unknown about the NaCl tolerance of coastal unicellular N$_2$-fixers, which have only recently been recognised for their significant contribution to the oceanic N cycle[1,7].

Here, we investigate the effect of NaCl on N$_2$ fixation using *Cyanothece* sp. ATCC 51142 (hereafter *Cyanothece* sp., recently reclassified as *Crocosphaera subtropica*[8]), a coastal N$_2$-fixing unicellular cyanobacterium[9]. First, we examine whether high salinity suppresses

[1]Shenzhen Public Platform for Screening and Application of Marine Microbial Resources, Tsinghua Shenzhen International Graduate School, Shenzhen, Guangdong Province, PR China. [2]National Innovation Center for Molecular Drug, Shenzhen, Guangdong Province, PR China. [3]School of Pharmaceutical Sciences, Tsinghua University, Beijing, PR China. [4]Institute of General Microbiology, Kiel University, Kiel, Germany. [5]Technology Innovation Center for Marine Ecology and Human Factor Assessment of Natural Resources Ministry, Tsinghua Shenzhen International Graduate School, Shenzhen, Guangdong Province, PR China. ✉e-mail: katrinhammerschmidt@googlemail.com; caizh@sz.tsinghua.edu.cn

unicellular cyanobacterial $N_2$ fixation, as is commonly proposed for multicellular cyanobacteria[4]. We then integrate physiological, transcriptomic, and enzymatic analyses to study the underlying mechanisms.

Contrary to the accepted view that high salinity negatively affects coastal $N_2$ fixation[4,5], here we demonstrate not only that $N_2$ fixation of *Cyanothece* sp. is NaCl-dependent, but rather that $Na^+$ energetics is essential to provide ATP for $N_2$ fixation. Overall, we offer an alternative perspective on the paradox of the scarcity of $N_2$-fixing cyanobacteria in N-limited coastal waters and provide insights into the role of $Na^+$ energetics in cyanobacterial metabolism, specifically in $N_2$ fixation.

## Results

### $N_2$ fixation depends on the presence of NaCl

We first carried out growth experiments of *Cyanothece* sp. cultured in artificial seawater medium (ASP2, 18 g/L NaCl) and freshwater medium (BG11) in the presence or absence of N. Under N-rich conditions, cells from ASP2 and BG11 showed similar population growth, as expected for a coastal isolate. Under N deficiency, population growth was only observed in ASP2 without N (hereafter ASP2-N), whereas cells barely grew in BG11 without N (hereafter $BG11_0$) (Fig. 1a). Since the most prominent discrepancy between ASP2-N and $BG11_0$ is the presence of NaCl, we recultured these non-growing cells in fresh $BG11_0$ amended with 18 g/L NaCl (hereafter $BG11_0$ (NaCl)), and NaCl reactivated population growth (Fig. 1b). Here $N_2$ fixation is the prerequisite for cell growth and division, it is therefore confirmed that $N_2$ fixation depends on NaCl. Notably, although a slight population growth of cells grown in $BG11_0$ was observed in the first few days after inoculation, this was probably due to the consumption of the intracellular nitrogen storage product cyanophycin and the degradation of some proteins (such as light-harvesting complex phycobilisomes)[10] rather than $N_2$ fixation, which would otherwise have resulted in continuous population growth.

The effect of NaCl gradients on $N_2$ fixation and growth was investigated in more detail. When N was replete, except for the three highest NaCl concentrations (32, 34, 36 g/L) (ANOVA, $F_{3,8} = 22.04$, $p < 0.001$), cells in the other treatments did not differ in growth from cells grown at 18 g/L, which is equivalent to the salinity of the marine medium ASP2 (Fig. 1c). In contrast, when N was depleted, NaCl was required for $N_2$ fixation and population growth. Cell proliferation was almost completely inhibited when NaCl was less than 4 g/L and maximised at 18 g/L NaCl (Fig. 1d, ANOVA, $F_{3,8} = 137.1$, $p < 0.001$). The requirement for NaCl on $N_2$ fixation was further verified by the direct quantification of $N_2$ fixation activities. In accordance with the growth pattern (Fig. 1d), $N_2$ fixation activities were almost completely inhibited at low NaCl concentrations (0, 2 g/L) and maximised at 18 g/L NaCl (approximately 75 nmol ethylene produced per $10^8$ cells per hour) (Fig. 1e, ANOVA, $F_{3,8} = 318.3$, $p < 0.001$). In addition, we found that NaCl is a general requirement for $N_2$ fixation irrespective of cellular growth stage, as cells from the exponential phase (Fig. 1a), the stationary phase (Supplementary Fig. 1), and non-$N_2$-fixing growth-arrested cells (Fig. 1b) all required NaCl to fix $N_2$ under diazotrophy.

### Cellular N starvation due to lack of $N_2$ fixation

In the absence of NaCl, non-$N_2$-fixing growth-arrested exhibited a yellow colouration (Supplementary Fig. 2a), a typical symptom of nitrogen-starvation-induced chlorosis[10]. To test whether these non-growing cells in $BG11_0$ are physiologically N-starved due to the non-functional $N_2$ fixation, nitrogenous compounds and dry weight of cells cultured in either $BG11_0$ or $BG11_0$ (NaCl) were quantified. Consistent with our expectation, compared to normally growing cells (functioning $N_2$ fixation) with NaCl, NaCl deprivation led to a significant

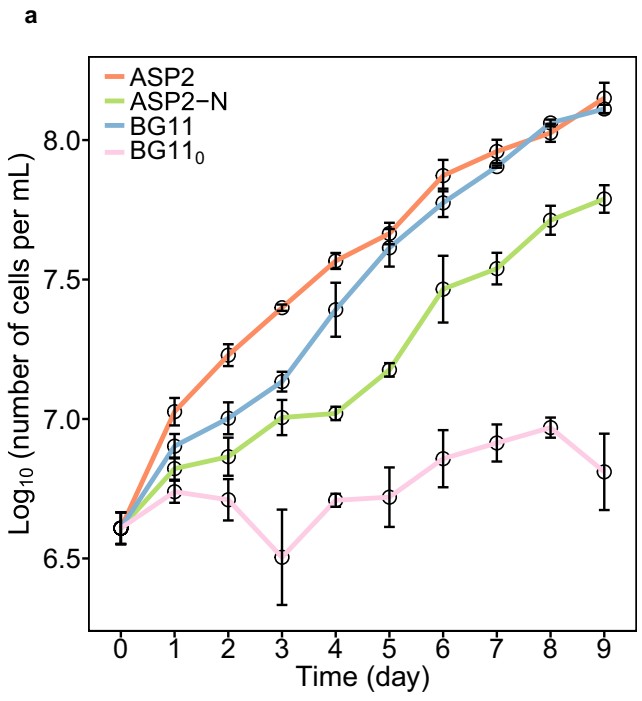

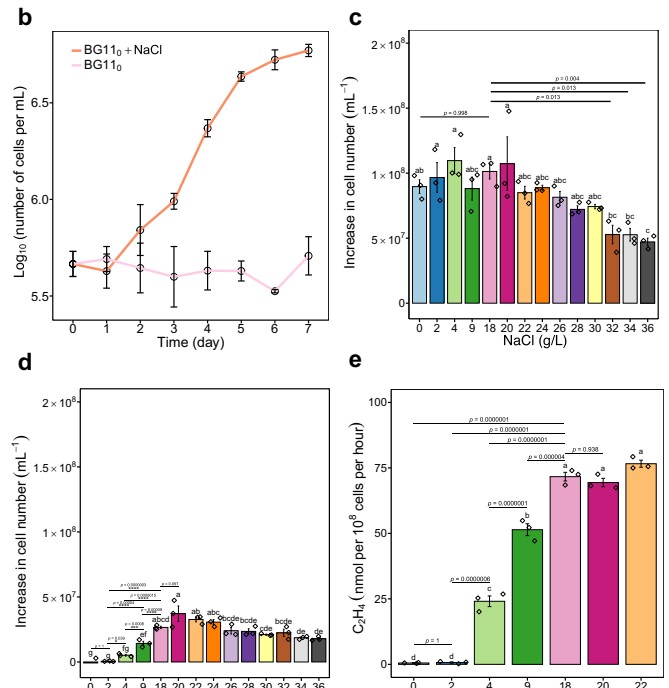

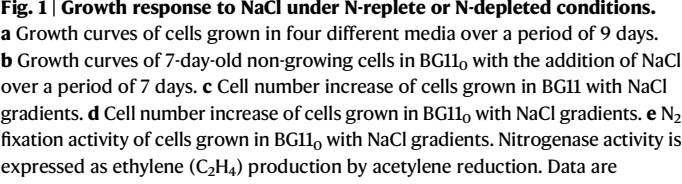

**Fig. 1 | Growth response to NaCl under N-replete or N-depleted conditions.** **a** Growth curves of cells grown in four different media over a period of 9 days. **b** Growth curves of 7-day-old non-growing cells in $BG11_0$ with the addition of NaCl over a period of 7 days. **c** Cell number increase of cells grown in BG11 with NaCl gradients. **d** Cell number increase of cells grown in $BG11_0$ with NaCl gradients. **e** $N_2$ fixation activity of cells grown in $BG11_0$ with NaCl gradients. Nitrogenase activity is expressed as ethylene ($C_2H_4$) production by acetylene reduction. Data are presented as mean ± standard deviation ($n = 3$ biologically independent samples). Different letters on each bar (**c**, **d**, **e**) indicate statistical significance ($p < 0.05$) calculated by one-way ANOVA with Tukey's HSD post-hoc analysis over all populations tested. In figure **d**, line segments and corresponding significance markers indicate statistical significance calculated by one-way ANOVA across four salinity treatments (2, 4, 9, 18 g/L). Significance: ns (no significance), *($p < 0.05$), **($p < 0.01$), ***($p < 0.001$), ****($p < 0.0001$). Source data are provided as a Source Data file.

decrease in chlorophyll ($C_{55}H_{72}O_5N_4Mg$) ($t$ test, $p < 0.05$), total protein content ($t$ test, $p < 0.01$), dry weight ($t$ test, $p < 0.01$), and total protein ratio (of dry weight) ($t$ test, $p < 0.05$) of non-$N_2$-fixing cells grown in BG11$_0$ (Supplementary Figs. 2b-e), indicating that these cells were N-starved due to the lack of $N_2$ fixation.

## Transcriptomic profiles in the presence and absence of NaCl

Transcriptomic analysis was performed to reveal the molecular background and transcriptomic features of the NaCl requirement for $N_2$ fixation. Compared to $N_2$-fixing cells cultured with NaCl (18 g/L), 5226 (1909 genes upregulated, 3317 genes downregulated) genes of non-$N_2$-fixing cells grown without NaCl were identified as differentially expressed genes (DEGs) (adjusted $P$ value < 0.05) (Supplementary Fig. 3a). According to KEGG enrichment analysis, all DEGs could be classified into several functional categories (Supplementary Fig. 3b), showing striking metabolic differences. Detailed heatmap visualisation of DEGs enriched in KEGG ribosome biosynthesis (map03010, Fig. 2a), photosynthesis (map00195, Fig. 2b), and porphyrin metabolism (participating in chlorophyll synthesis) (map00860, Supplementary Fig. 3c) indicated that almost all DEGs enriched in non-$N_2$-fixing cells showed a downregulated pattern at the transcript level compared to $N_2$-fixing cells. Given the importance of ribosomes and photosynthesis for cell growth and division, these distinct gene expression patterns provide clues at the molecular level as to why diazotrophic cells cultured without NaCl fail to grow or divide.

It is reasonable to expect that non-$N_2$-fixing (N-starved) cells would reduce their primary metabolism to a minimal level in order to survive. Indeed, these cells exhibited the stringent response[11] (a persistent state in which cells reduce protein synthesis and stop dividing), as evidenced by non-$N_2$-fixing cells in the absence of Na$^+$: 1) Cells were nitrogen-starved (Supplementary Fig. 2) and growth-arrested (Figs. 1a, d), which represents the most typical cause and consequence of the stringent response[11,12]; 2) They exhibited typical stringent response transcriptomic patterns, which involved the downregulation of ribosome biosynthesis genes (Fig. 2a) and the upregulation of nutrient transporter genes[12] (Supplementary Fig. 4a, 32 out of 45 annotated genes); 3) They upregulated the expression of the *relA* gene (Supplementary Fig. 4b, gene location: NC_010546.1 (21413–22405), NCBI), encoding ppGpp synthase/hydrolase, a key enzyme that regulates levels of ppGpp (the signalling molecule for the stringent response)[11]. Intriguingly, although *Cyanothece* sp. is capable of $N_2$ fixation, the morphological, physiological and transcriptomic performance of the non-$N_2$-fixing growth-arrested cells in the absence of Na$^+$ also align with the nitrogen chlorosis theory, which typically describes the processes involved in the acclimation of non-diazotrophic cyanobacteria to N shortage[10]. It is important to note, however, that the stringent response (or nitrogen chlorosis) described above is the consequence (resulting from cellular N starvation) rather than the cause of $N_2$ fixation dysfunction in the absence of NaCl. The reason for the $N_2$-fixers' N-starvation in the absence of NaCl has yet to be determined.

With regard to the nitrogenase (the key enzyme responsible for $N_2$ fixation) structural genes (*nifHDK*)[13], we found a higher level of expression of *nifHDK* in the non-$N_2$-fixing cells, i.e., cultured without NaCl, than in the $N_2$-fixing cells, cultured with NaCl (Fig. 2c). This expression pattern suggests that non-$N_2$-fixing growth-arrested cells in the absence of Na$^+$ were N-starved, which is in accordance with the result that these cells were physiologically N-starved (Supplementary Fig. 2).

## $N_2$ fixation depends exclusively on Na$^+$

To determine whether the need for NaCl is specific, we tested whether four other metal chlorides, including KCl, LiCl, $MgCl_2$ and $CaCl_2$, at doses equivalent to NaCl (310 mM, following the medium recipe for ASP2[9]), could stimulate the growth of cells grown in BG11$_0$ in the same way as NaCl. As $N_2$ fixation is a must for population growth under N deprivation, population growth is used as a proxy for $N_2$ fixation activity in the following contexts, unless otherwise specified. We found that population growth was exclusively observed in the NaCl treatment (Fig. 3a, ANOVA, $F_{4,10} = 65.74$, $p < 0.001$). Given that high concentrations of LiCl are toxic to cells and the observed growth arrest of the LiCl (310 mM) treatment could have been caused by its toxicity, we further tested whether LiCl at low concentrations can activate population growth. The results show that LiCl at low concentrations cannot stimulate population growth either (Supplementary Fig. 5), indicating that LiCl cannot function in the same manner as NaCl to activate $N_2$ fixation. Overall, these results demonstrate that the requirement for NaCl for $N_2$ fixation is exclusive and cannot be substituted by other metal chlorides. Moreover, since all the tested metal chlorides contain a chloride moiety, this result also indicates that Na$^+$ and not Cl$^-$ plays a role in $N_2$ fixation. Notably, BG11$_0$ medium alone contains 0.082 g/L Na$^+$ from its constitutive ingredients ($Na_2CO_3$, $Na_2EDTA \cdot 2H_2O$ and $Na_2MoO_4 \cdot 2H_2O$), nevertheless, this Na$^+$ concentration is far below the effective concentration (4 g/L) for activating $N_2$ fixation as shown earlier (Figs. 1d, e).

## Active nitrogenase despite the lack of Na$^+$

Unicellular cyanobacterial $N_2$ fixation can generally be divided into two steps, i.e., $N_2$ fixation (from gaseous $N_2$ to ammonium) and N assimilation (from ammonium to organic nitrogenous compounds)[14] (Fig. 3b). We have shown that $N_2$ fixation is severely inhibited (Fig. 1e), but it remains to be determined whether N assimilation is also suppressed under NaCl deprivation. Therefore, we performed a nutrient addition experiment in which the product of $N_2$ fixation (ammonium ($NH_4Cl$)) and an intermediate of N assimilation (glutamine (Gln)) was added to cells grown in BG11$_0$. Compared to cells grown in BG11$_0$, the addition of $NH_4Cl$ (2 mM) and Gln (2 mM) mimicked the Na$^+$ effect and resulted in normal growth (Fig. 3c, ANOVA, $F_{3,8} = 58.83$, $p < 0.001$), confirming that $N_2$ fixation only was inhibited by Na$^+$ deprivation.

We then tested: 1) whether Na$^+$ deficiency inhibits the biosynthesis of nitrogenase; 2) whether Na$^+$ plays a role in protecting the enzyme from oxygen ($O_2$) inactivation, as nitrogenase is known to be extremely sensitive to $O_2$[15]; 3) whether Na$^+$ participates in the conformational activation of nitrogenase. Although almost no $N_2$ fixation activity was detected in cells grown in BG11$_0$ (Fig. 1e), these cells expressed a significantly higher amount of nitrogenase (Fig. 3d, ANOVA, $F_{2,15} = 337.6$, $p < 0.001$), which is consistent with our previous transcriptomic analysis showing that non-$N_2$-fixing cells expressed higher levels of *nifHDK* (Fig. 2c). The results of the sparging experiment show that even under anaerobic conditions, Na$^+$ was still required for $N_2$ fixation and population growth (Fig. 3e, ANOVA, $F_{3,8} = 40.52$, $p < 0.001$), suggesting that the dependence of nitrogenase activity on Na$^+$ is not due to a role of the cation in providing oxygen protection to the enzyme. Nitrogenase from non-$N_2$-fixing cells was tested to be conformationally active with a significantly higher catalytic rate compared to that of $N_2$-fixing cells (Fig. 3f, ANOVA, $F_{2,15} = 289.2$, $p < 0.001$), which makes sense since the catalytic capacity of an enzyme is generally proportional to its amount (Fig. 3d). Taken together, these results indicate that: 1) Na$^+$ is not involved in the biosynthesis, protection against $O_2$ inactivation or conformational activation of nitrogenase; 2) non-$N_2$-fixing cells grown without Na$^+$ synthesised an unusually high amount of nitrogenase but could not fix $N_2$.

## A shortage of ATP prevents $N_2$ fixation

Due to the incompatibility of oxygen-producing photosynthesis and $O_2$-sensitive $N_2$ fixation, $N_2$ fixation of *Cyanothece* sp. occurs in the dark[16,17]. It is well established that $N_2$-fixing cells synthesise and store large quantities of carbohydrate in the form of glycogen via photosynthesis in the light, which is used to generate adenosine triphosphate (ATP) via glycolysis to power $N_2$ fixation in the dark[18]. Given that

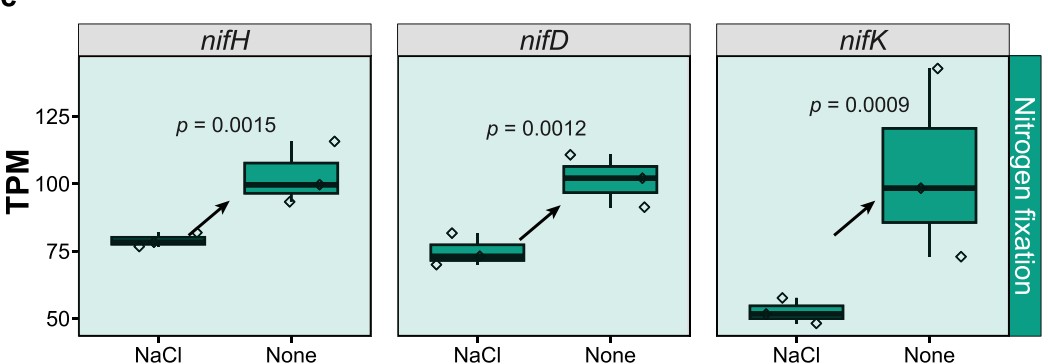

**Fig. 2 | Enrichment analysis of ribosome biosynthesis and photosynthesis genes and gene expression trend of nitrogenase. a** Heatmap analysis showing the DEGs enriched for ribosome biosynthesis (map03010) and (**b**) for photosynthesis (map00195). Two components of the ribosomal proteins (small subunit, large subunit) and six photosynthetic subunits (photosystem II (PSII), cytochrome, photosystem I (PSI), ATP synthase, allophycocyanin, phycocyanin) were analysed. KEGG annotations were assigned from the genome annotation. Column dendrograms indicate similarity based on Euclidean distance and hierarchical clustering. Gene clusters were determined by k-means clustering with Euclidean distance. The heatmap colour gradient shows low gene expression (blue) and high gene expression (red). **c** Expression trend of nitrogenase structural genes (*nifHDK*). Data

(TPM, transcripts per million) are presented as box plots (lower bound at 25th percentile, centre line at the median, upper bound at 75th percentile) with whiskers at minimum and maximum values. Shaded-in plots indicate significantly different expression levels of genes and arrows indicate the directionality of statistically significant trends between these two treatments ($p < 0.05$, right-tailed Fisher's exact $t$ test followed by Benjamin-Hochberg (BH) adjustment). Statistical significance of selected genes was calculated from pairwise TPM comparisons of triplicate samples. "NaCl" denotes the addition of extra NaCl (18 g/L) to BG11$_0$, while "None" indicates that no additional substances were added to BG11$_0$. Source data are provided as a Source Data file.

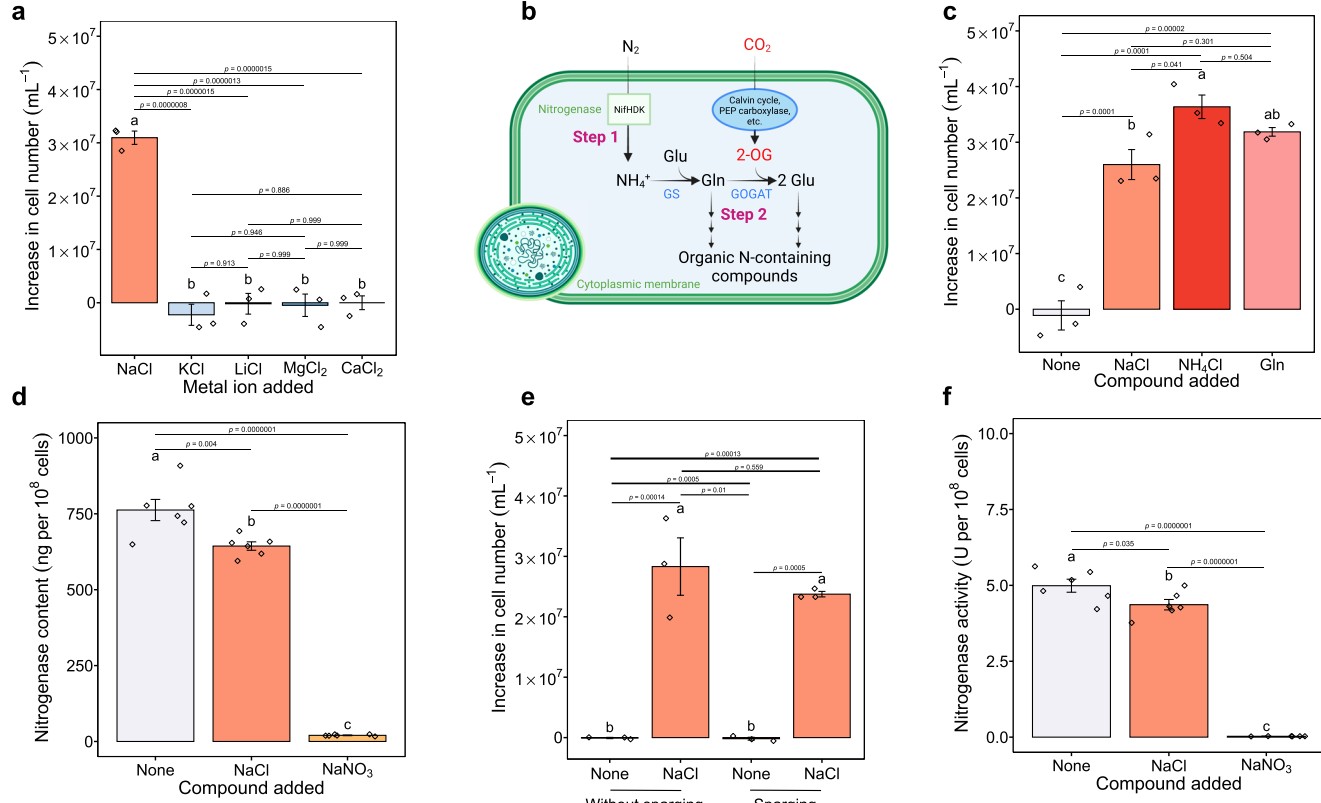

**Fig. 3 | Possible mechanisms underlying the NaCl requirement for $N_2$ fixation.**
**a** Increase in cell number of cells grown in $BG11_0$ with the addition of five different metal chlorides: NaCl, KCl, LiCl, $MgCl_2$ and $CaCl_2$. **b** Schematic representation of the pathway of unicellular cyanobacterial $N_2$ fixation. Created in BioRender. Tang, S. (2023) BioRender.com/e77x697. $N_2$ is first reduced to $NH_4^+$ by nitrogenase (Step 1) and then is incorporated into carbon skeletons via the glutamine (Gln) synthetase-glutamate (Glu) synthase pathway, where further N-containing organic metabolites are formed (Step 2). NifHDK, nitrogenase complex; PEP carboxylase, phosphoe-nolpyruvate carboxylase; 2-OG, 2-oxoglutarate; GS, glutamine synthetase; GOGAT, glutamate synthase. **c** Increase in cell number of cells grown in $BG11_0$ amended with NaCl, $NH_4Cl$ and Gln. **d** Nitrogenase content of cells grown in $BG11_0$, $BG11_0$ (NaCl) and $BG11_0$ (NaNO₃). **e** Cell number increase of cells grown in $BG11_0$ and $BG11_0$ (NaCl) with and without $N_2$ sparging. **f** Nitrogenase activity of cells grown in $BG11_0$, $BG11_0$ (NaCl) and $BG11_0$ (NaNO₃). One unit (U) is the amount of enzyme that catalyses the reaction of 1 μmol of substrate per minute. The graphs show the mean ± standard deviation, $n = 3$ biologically independent samples for **a**, **c** and **e**, $n = 6$ biologically independent samples for (**d** and **f**). Different letters on each bar represent statistical significance ($p < 0.05$) calculated by one-way ANOVA with Tukey's HSD post-hoc analysis over all populations tested. Source data are provided as a Source Data file.

biological $N_2$ fixation, including both $N_2$ fixation itself (the theoretical minimum energy requirement) and the associated apparatuses supporting $N_2$ fixation, is highly energy-intensive[16], we reason that a limitation of ATP caused by $Na^+$ deficiency might lead to $N_2$ fixation dysfunction.

It has been reported that $Na^+$ plays a role in bicarbonate uptake, which is subsequently catalysed to release $CO_2$ for photosynthesis[19,20]. Therefore, $Na^+$ deprivation may adversely affect photosynthesis and glycogen biosynthesis, resulting in glycogen shortage (the energy substance for $N_2$ fixation) and insufficient ATP supply for $N_2$ fixation. Accordingly, we determined whether $Na^+$ was involved in glycogen synthesis and indirectly inhibited $N_2$ fixation. To this end, we quantified and contrasted the intracellular glycogen levels of cells grown without N, either in the presence or absence of $Na^+$, under ambient $CO_2$ levels without forced aeration, in a diurnal cycle (a light-dark cycle of 12 h/12 h). As expected, the cellular glycogen content of $N_2$-fixing cells grown in $BG11_0$ (NaCl) exhibited a diurnal pattern, with glycogen synthesis occurring during the light period and consumption to power $N_2$ fixation in the dark period. It is noteworthy that, in comparison to $N_2$-fixing cells, non-$N_2$-fixing growth-arrested cells in $BG11_0$ exhibited elevated levels of glycogen, yet were unable to utilise it in the dark (Fig. 4a, ANOVA, $F_{3,8} = 37.36$, $p < 0.001$). Indeed, the accumulation of glycogen in cyanobacteria under N starvation represents a classic physiological response observed during nitrogen chlorosis[10]. This result, coupled with the fact that cells can grow normally in BG11

(Fig. 1a), suggests that $Na^+$ is not involved in bicarbonate uptake and glycogen synthesis and that the energy source (glycogen) shortage is not the cause of $N_2$ fixation failure in the absence of $Na^+$.

To further test the hypothesis that an ATP shortage in the absence of $Na^+$ leads to $N_2$ fixation dysfunction, an additional supply of ATP was provided to cells grown in $BG11_0$. The results show that additional ATP (0.1 mM) stimulated population growth (Fig. 4a, ANOVA, $F_{2,6} = 49.95$, $p < 0.001$), suggesting that: 1) The observed failure of the nitrogenase to function may be attributed to the lack of ATP; 2) Under diazotrophic conditions, $Na^+$ may play a role in ATP biosynthesis. However, the presence of the nitrogen element in ATP ($C_{10}H_{16}N_5O_{13}P_3$) promoted us to reconsider the role of ATP in activating $N_2$ fixation, either by providing chemical energy or by providing N. To distinguish between these two possibilities, we first examined the nitrogenase content of cells grown in $BG11_0$ supplemented with additional ATP. Almost no nitrogenase was observed in cells supplemented with N (Gln (0.1 mM) and NaNO₃ (0.1 mM)), whereas cells supplemented with ATP (0.1 mM) or $Na^+$ (310 mM) synthesised the two highest levels of nitrogenase (Fig. 4b, ANOVA, $F_{3,20} = 2269$, $p < 0.001$). It can be concluded that in this case ATP does not act as a N supplier, which would otherwise not lead to the biosynthesis of nitrogenase. Furthermore, we also assessed population growth in $BG11_0$ supplemented with an equivalent dose (0.1 mM) of ATP, ADP (adenosine diphosphate, $C_{10}H_{15}N_5O_{10}P_2$), or AMP (adenosine monophosphate, $C_{10}H_{14}N_5O_7P$). As shown in Fig. 4c, cells supplemented with ATP had the highest growth among all treatments

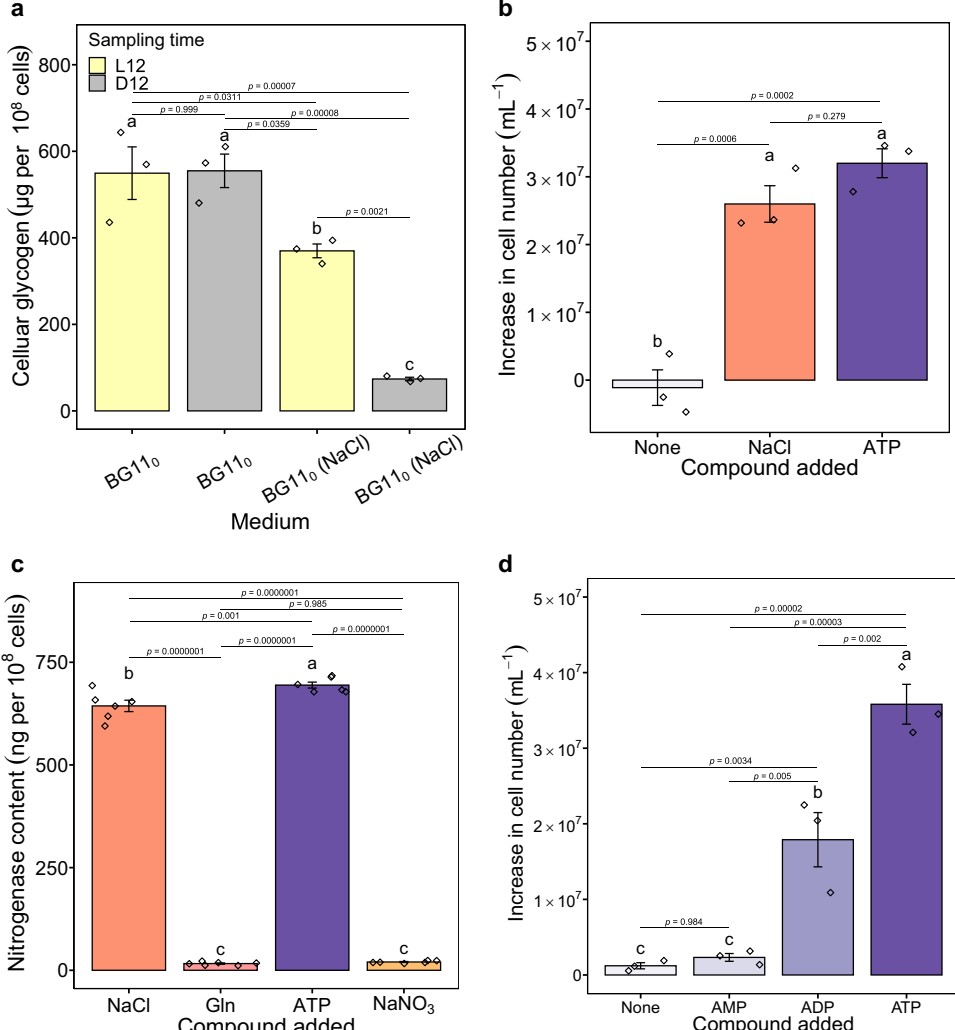

**Fig. 4 | ATP shortage underlies the dysfunction of $N_2$ fixation in the absence of $Na^+$. a** Cellular glycogen contents of non-$N_2$-fixing cells grown in $BG11_0$ or $N_2$-fixing cells grown in $BG11_0$ with 18 g/L NaCl ($BG11_0$ (NaCl)) in a diurnal cycle. Cells were sampled at the end of the light phase (L12) or at the end of the dark phase (D12). **b** Increase in cell number of cells grown in $BG11_0$ or supplemented with either NaCl or ATP. **c** Nitrogenase content of cells grown in $BG11_0$ (NaCl), $BG11_0$ (Gln), $BG11_0$ (ATP) and $BG11_0$ (NaNO_3). **d** Increase in cell number of cells grown in $BG11_0$

supplemented with equivalent doses of ATP, ADP or AMP. The graphs show the mean ± standard deviation, $n = 3$ biologically independent samples for **a**, **b** and **d**, $n = 6$ biologically independent samples for (**c**). Different letters on each bar represent statistical significance ($p < 0.05$) calculated by ANOVA with Tukey's HSD post-hoc analysis across all populations tested. Source data are provided as a Source Data file.

(ANOVA, $F_{3,8} = 52.06$, $p < 0.001$), indicating that ATP serves as additional chemical energy rather than as an N source. Otherwise, this would have resulted in comparable growth patterns across all treatments, as all supplements provided an equivalent dose of N. From an energetic point of view, this result makes sense, as ATP possesses a higher energy content than ADP and AMP, thereby enabling the nitrogenase to fix more $N_2$ for growth. Notably, although not statistically significant, there was a slight population growth in the AMP treatment when compared to the no additive control. This is probably due to the scavenging of the very low energy that AMP contains. Although it has been reported that many bacterial species are capable of releasing and actively depleting extracellular ATP[21], further investigation is required to ascertain how *Cyanothece* sp. cells utilise extracellular ATP. Possibilities include that ATP may be transported into cells via water-filled porins (at least into the periplasm)[22,23] or via transporters/carriers[24] or other uptake systems, either independently or in combination, and then utilised by cells. In conclusion, the results collectively indicate that insufficient bioavailable ATP (energy) impairs $N_2$ fixation in the absence of $Na^+$. Nevertheless, the mechanism by

which $Na^+$ contributes to the energy metabolism of $N_2$-fixing cells remains elusive.

## $Na^+$ energetics enables $N_2$ fixation

Regarding energy metabolism in bacteria, it is well established that ATP synthesis is driven by the energy stored in a transmembrane electrochemical gradient of protons coupled to the light-driven or the respiration-driven proton pumps ($H^+$ energetics)[25]. Nevertheless, there is evidence that $Na^+$ can replace $H^+$ as the coupling ion to drive ATP synthesis in several anaerobic bacteria[25,26], as well as in cyanobacteria, as recently demonstrated[20]. This type of machinery for synthesising ATP by $Na^+$-coupling ($Na^+$ energetics) fits well with our findings that $Na^+$ correlates with the supply of ATP for $N_2$ fixation. Also, there is a striking resemblance between $N_2$-fixing *Cyanothece* sp. and anaerobic bacteria with $Na^+$ energetics, as both require non-oxic microenvironments. Therefore, we propose that a mechanism for $Na^+$-coupled ATP synthesis exists in $N_2$-fixing *Cyanothece* sp. To test this proposal, we first quantified the cellular ATP content and the ATP/ADP ratio of non-$N_2$-fixing cells grown in $BG11_0$ (for seven days) following the addition of

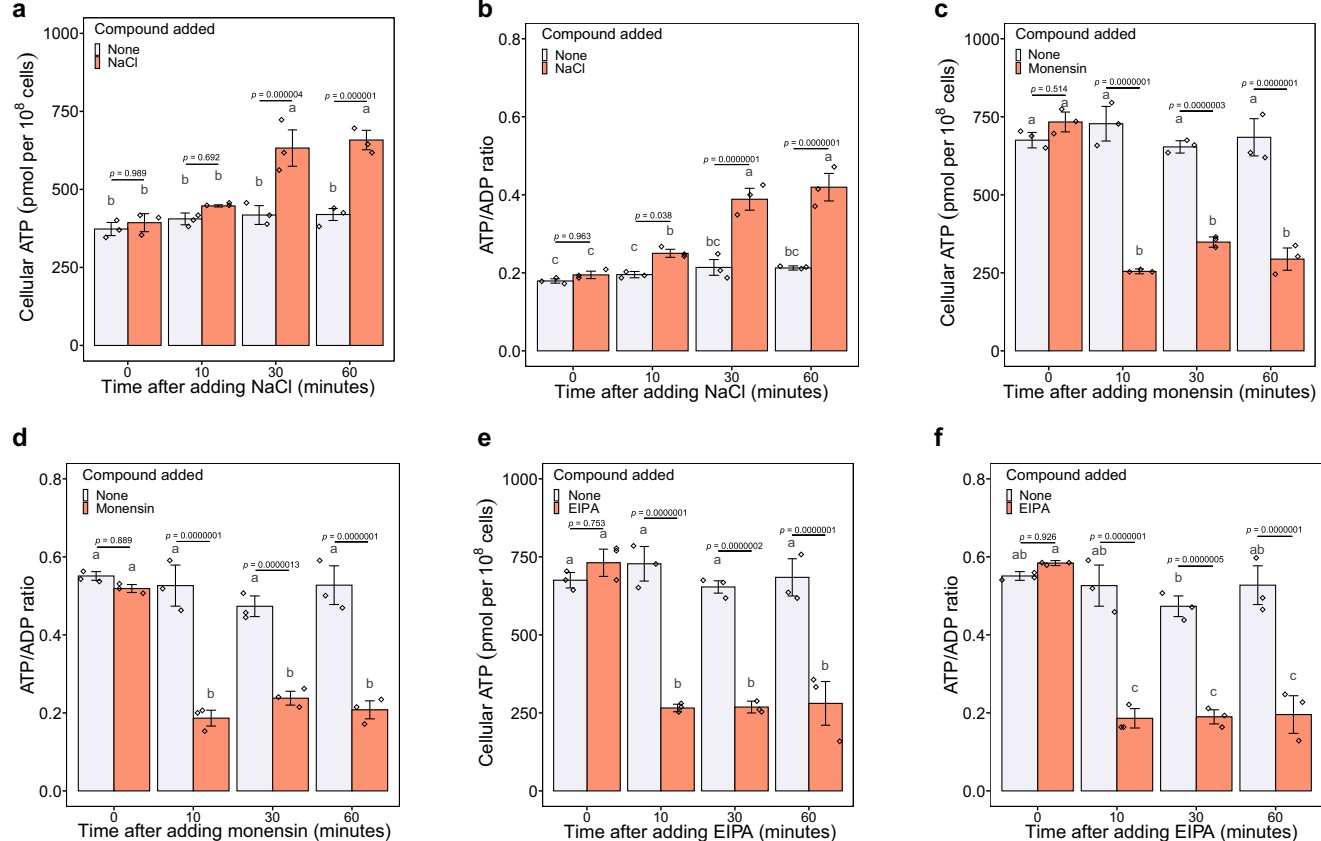

**Fig. 5 | Cellular ATP levels and ATP/ADP ratios of $N_2$-fixing cells in the presence of NaCl, monensin or EIPA.** Cellular ATP content (**a**) and ATP/ADP ratio (**b**) of N-starved cells grown in $BG11_0$ after the addition of NaCl in the dark. Prior to NaCl addition, cells were grown in $BG11_0$ for 7 days. Cellular ATP content (**c**) and ATP/ADP ratio (**d**) of dark phase $N_2$-fixing cells grown in $BG11_0$ with 18 g/L NaCl after the addition of monensin (14 μM). Cellular ATP content (**e**) and ATP/ADP ratio (**f**) of dark phase $N_2$-fixing cells grown in $BG11_0$ with 18 g/L NaCl after the addition of EIPA (100 μM). The graphs show the mean ± standard deviation ($n = 3$ biologically independent samples). Different letters on each bar represent statistical significance ($p < 0.05$) calculated by ANOVA with Tukey's HSD post-hoc analysis across all populations tested. Source data are provided as a Source Data file.

NaCl (18 g/L) during $N_2$ fixation in the dark. The addition of NaCl resulted in a significantly higher amount of ATP (Fig. 5a, ANOVA, $F_{3,16} = 21.28$, $p < 0.001$) and higher ATP/ADP ratio (Fig. 5b, ANOVA, $F_{3,16} = 37.02$, $p < 0.001$) relative to the no NaCl addition control, indicating that $Na^+$ plays a critical role in driving ATP synthesis. To further corroborate the involvement of $Na^+$ in ATP synthesis during $N_2$ fixation, dark-phase $N_2$-fixing cells grown with NaCl were treated with monensin[20,27] (a $Na^+$-specific ionophore) or ethyl-isopropyl amiloride[20] (EIPA, an inhibitor of $Na^+$ channels and $Na^+/H^+$ antiporters), respectively. The ionophore monensin renders membranes permeable to $Na^+$ and thus destroys transmembrane $Na^+$ gradients, whereas EIPA suppresses transmembrane $Na^+$ transportation. If $Na^+$ fluxes were coupled to ATP synthesis, the collapse of the $Na^+$ transmembrane gradient caused by monensin or EIPA should result in a reduction in cellular ATP levels and ATP/ADP ratios due to the inhibition of ATP synthesis. As shown in Figs. 5c, d, the addition of monensin (14 μM) resulted in significantly lower ATP levels (ANOVA, $F_{3,16} = 64.21$, $p < 0.001$) and ATP/ADP ratios (ANOVA, $F_{3,16} = 31.87$, $p < 0.001$) in comparison to the untreated control. Similarly, treatment with EIPA (100 μM) markedly suppressed ATP synthesis (Fig. 5e, ANOVA, $F_{3,16} = 45.72$, $p < 0.001$), leading to lower ATP/ADP ratios (Fig. 5f, ANOVA, $F_{3,16} = 40.3$, $p < 0.001$). These results substantiate the pivotal role of $Na^+$ energetics in $N_2$ fixation.

To further understand the role of $Na^+$ energetics in energy metabolism, we investigated cell growth after adding either monensin, a protonophore 3,5-di-tert-butyl4-hydroxybenzaldehyde (DTHB)[28], EIPA, or a typical F-type ATP synthase inhibitor dicyclohexylcarbodiimide

(DCCD)[29] to the respective culture medium. Similar to monensin that destroys $Na^+$ energetics, the protonophore DTHB makes membranes permeable to protons, abolishing proton gradients and thus inhibiting $H^+$ energetics and ATP generation. Compared to normally growing cells from the control, 14 μM monensin completely inhibited the growth of cells in $BG11_0$ (NaCl) (Fig. 6a, ANOVA, $F_{3,16} = 135.2$, $p < 0.001$). Conversely, cells cultured in BG11 with monensin divided normally, indicating that monensin is not toxic to cells. Furthermore, cells were cultured in $BG11_0$ supplemented with ATP in the presence or absence of monensin, to examine whether monensin inhibits the performance of nitrogenase. The result shows that the addition of monensin does not impede the growth of ATP-supplemented cells, indicating that monensin does not disrupt the nitrogenase (Supplementary Fig. 6). With regard to DTHB exposure, only the cells that had been grown in BG11 were significantly inhibited by DTHB (200 μM) (Fig. 6b, ANOVA, $F_{3,16} = 169.83$, $p < 0.001$). No growth was observed in any of the treatments that were exposed to EIPA (100 μM) (Fig. 6c, ANOVA, $F_{3,16} = 99.11$, $p < 0.001$) or DCCD (100 μM) (Fig. 6d, ANOVA, $F_{3,16} = 249.3$, $p < 0.001$). These results suggest: 1) The presence of a $Na^+$-coupled ATP synthesis machinery. This was demonstrated by non-toxicity of monensin to cells and nitrogenase, as well as by the abolition of transmembrane $Na^+$ gradients, which resulted in the inhibition of ATP synthesis and the subsequent growth arrest of cells in $BG11_0$ (NaCl); 2) The existence of an $H^+$-coupled ATP synthesis machinery was evidenced by the normal growth of cells in BG11, which was unaffected by monensin, but was severely impaired by DTHB. Furthermore, we found that cells in $BG11_0$ (NaCl) with DTHB grew normally, suggesting that $H^+$ energetics is not involved in $N_2$ fixation; 3) The inhibition of the

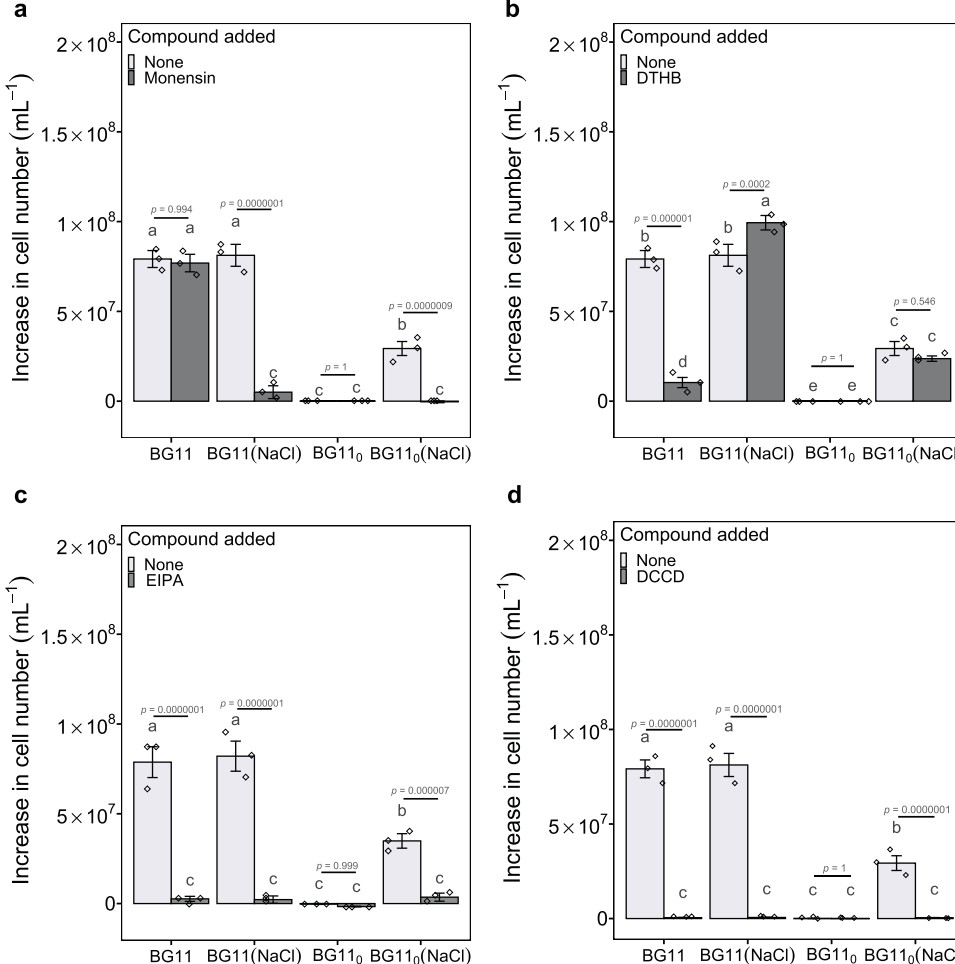

**Fig. 6 | Population growth with monensin, DTHB, EIPA or DCCD.** Increase in number of cells grown in the medium supplemented with monensin (**a**), DTHB (**b**), EIPA (**c**) or DCCD (**d**) as compared to non-supplemented cells. The graphs show the mean ± standard deviation ($n$ = 3 biologically independent samples). Different letters on each bar represent statistical significance ($p < 0.05$) calculated by ANOVA with Tukey's HSD post-hoc analysis across all populations tested. Source data are provided as a Source Data file.

Na$^+$/H$^+$ antiporter and suppression of other essential cellular functions through EIPA, as evidenced by the complete suppression of cell growth in all treatments. Indeed, EIPA has been reported to severely inhibit cyanobacterial PSII system[20]. This suggests that lower cellular ATP contents and ATP/ADP ratios observed following EIPA treatment (Fig. 5e, f) may be attributed not only to the inhibition of Na$^+$-dependent ATP synthesis but also to the suppression of other key metabolic processes; and 4) The utilization of the same type of F$_0$F$_1$ ATP synthase by Na$^+$ and H$^+$ energetics, as DCCD suppressed the growth of cells in all treatments. Interestingly, we found that cells in BG11 (NaCl) with monensin were unable to grow. This suggests that Na$^+$ availability may play a critical role in determining the coupling ion for ATP synthase. In this case, Na$^+$-coupled ATP synthesis would be favoured by the presence of Na$^+$, otherwise, the cells in BG11 (NaCl) with monensin should have grown due to H$^+$ energetics. Overall, these results indicate that Na$^+$ energetics plays an essential role in the activation of N$_2$ fixation. Nevertheless, future experimental efforts are needed to provide more direct evidence to determine whether and to what extent Na$^+$-dependent ATP synthesis takes place.

**Potential Na$^+$ gradient generator**
Similar to the H$^+$ cycle, a Na$^+$ cycle is required for continuous energy supply[25,26]. In this case, Na$^+$ pumps (Na$^+$ gradient generators) must

work together with ATP synthase (Na$^+$ gradient consumer) to re-energise the membrane in N$_2$-fixing cells. Since it has been reported that the Na$^+$/H$^+$ antiporter acts as a Na$^+$ pump to generate a trans-membrane Na$^+$ gradient[20], we examined the Na$^+$/H$^+$ antiporter of the respective cells. Using transcriptomic analysis, we found that in the absence of Na$^+$, non-N$_2$-fixing cells expressed significantly higher levels of four genes encoding subunits of the Na$^+$/H$^+$ antiporter, namely *mnhB* (K05566, KEGG), *mnhD* (K05568, KEGG), *mnhE* (K05569, KEGG), and *mnhG* (K05571, KEGG), than N$_2$-fixing cells (Supplementary Fig. 7), implying the key role of the Na$^+$/H$^+$ antiporter in generating the Na$^+$ gradient. Here, these cells were unable to establish a Na$^+$ gradient to drive ATP synthesis to meet their energy needs. Moreover, the result that N$_2$-fixing cells were extremely sensitive to EIPA (an inhibitor of the Na$^+$/H$^+$ antiporter, Fig. 6c), provides additional evidence that the Na$^+$/H$^+$ antiporter is involved in the regeneration of Na$^+$ gradients.

In summary, based on all the evidence provided in this study, we propose that *Cyanothece* sp. performs typical H$^+$ energetics across the thylakoid membrane during the day (Fig. 7a). In contrast, during the night under N$_2$-fixing conditions, we propose that the primary Na$^+$ gradient consumer (Na$^+$-coupled ATP synthase) works together with active Na$^+$ gradient generators (Na$^+$/H$^+$ antiporter) across the plasma membrane to drive ATP synthesis for N$_2$ fixation (Fig. 7b).

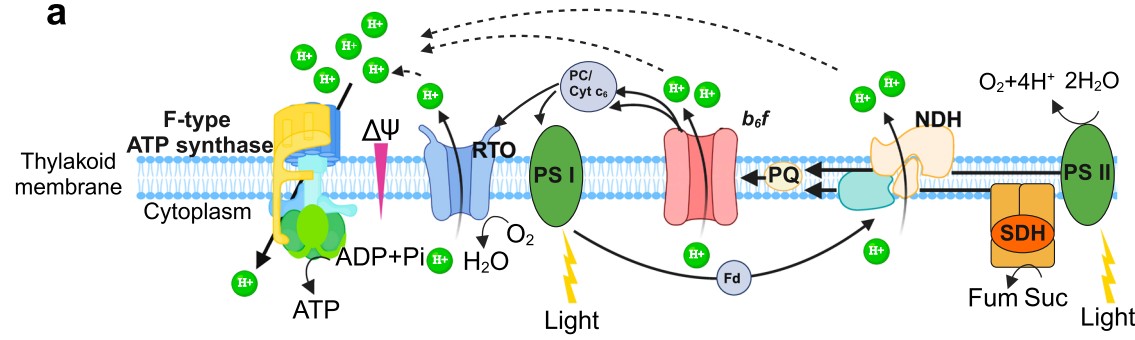

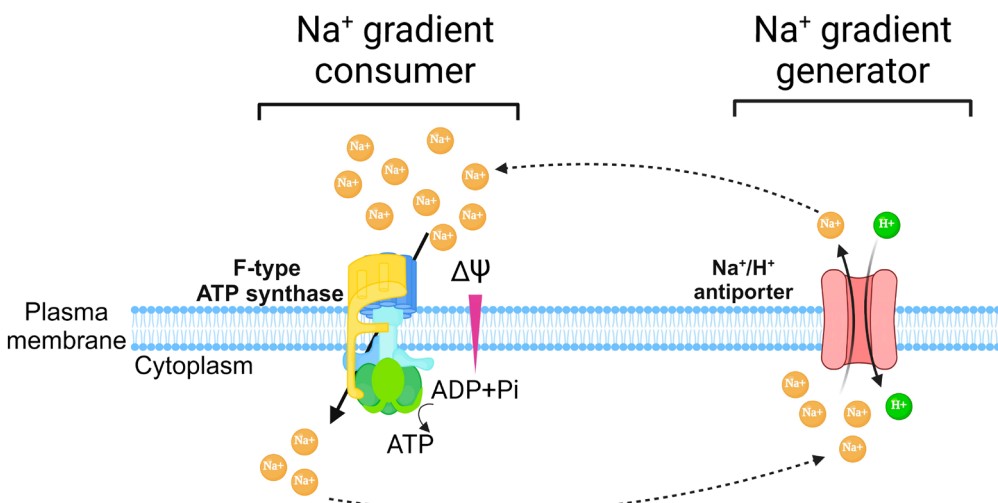

**Fig. 7 | Schematic illustration of proposed H⁺ and Na⁺ energetics in** ***Cyanothece*** **sp. a** Schematic representation of the typical H⁺ energetics across the thylakoid membrane coupled to photosynthesis in cyanobacteria (including *Cyanothece* sp.). Created in BioRender. Tang, S. (2023) BioRender.com/n32k956. **b** Schematic representation of the proposed Na⁺ energetics across the plasma membrane with Na⁺ gradient generator and consumer in N₂-fixing *Cyanothece* sp. cells. Created in BioRender. Tang, S. (2023) BioRender.com/n32k956.

Abbreviations: RTO, respiratory terminal oxidase; Cyt $b_6f$, cytochrome $b_6f$; PC/Cyt $c_6$, plastocyanin/cytochrome $c_6$; PQ, plastoquinone; Fd, flavodiiron; NDH, NADH-dehydrogenase; Fum, fumarate; Suc, succinate; SDH, succinate dehydrogenase; Glu, glutamic acid; Gln, glutamine; Ser, Serine; Val, valine; Tyr, tyrosine; Ala, alanine; ATP, Adenosine triphosphate; ADP, Adenosine diphosphate; Pie, inorganic phosphate; ΔΨ, membrane potential.

## Anaerobic fermentation possibly powers N₂ fixation

To understand why *Cyanothece* sp. cells capable of H⁺ energetics must perform Na⁺ energetics under N₂-fixing conditions, it is important to note that N₂ fixation in this species occurs in the dark and under anaerobic conditions in the cell[16]. Moreover, although H⁺ energetics are generally believed to be more energetically advantageous than Na⁺ energetics, H⁺ energetics in our case is no better than Na⁺ energetics due to the lack of a high potential electron acceptor, i.e., O₂[26]. Therefore, we propose that anaerobic energy metabolism coupled with Na⁺ energetics powers N₂ fixation in the case of *Cyanothece* sp.

Cellular O₂ dynamics during N₂ fixation of *Cyanothece* sp. under aerobic conditions show that although there is a sharp decrease in the O₂ level of the culture medium suggesting active aerobic respiration, this only occurs at the beginning of the dark phase[30,31]. In contrast, it has been repeatedly reported that the O₂ level remains almost unchanged, albeit with some perturbations (meaning that almost no O₂ is consumed by the cells), while at the same time active N₂ fixation has been detected during the remaining hours (indicating functioning energy metabolisms)[30,31]. This suggests that anaerobic energy metabolism, rather than aerobic respiration, is functioning to power N₂

fixation. Indeed, it has been shown that diazotrophic cyanobacteria (filamentous *Oscillatoria limosa* strain 23 and unicellular *Synechococcus* OS-A and OS-B') rely on anaerobic fermentation to power N₂ fixation under dark and anaerobic conditions[32,33]. Notably, *Cyanothece* sp. contains all the fermentation-related genes necessary to produce ethanol, lactate, and acetate in its genome[17], indicating the capacity for anaerobic fermentation. Taken together, this evidence suggests that it is highly likely that anaerobic fermentation plays a crucial role in powering N₂ fixation in *Cyanothece* sp.

To investigate whether fermentation powers N₂ fixation in *Cyanothece* sp., we first examined the transcriptomic level of key fermentation-related genes. Although our transcriptomic analysis was designed to discriminate non-N₂-fixing growth-arrested cells from normally growing N₂-fixing cells at the transcript level, we identified transcriptomic patterns of fermentation-related genes suggesting that fermentation is involved in N₂ fixation. Specifically, we found that three key genes involved in microbial fermentation[34], namely *ack* encoding acetate kinase (K00925, KEGG), *adh* encoding alcohol dehydrogenase (K00001, KEGG) and *ldh* encoding lactate dehydrogenase (K00016, KEGG), were upregulated in non-N₂-fixing

cells compared to $N_2$-fixing cells (Supplementary Fig. 8a). This suggests that non-$N_2$-fixing cells may be challenged to meet their energy requirements in the absence of $Na^+$ and therefore express higher levels of key fermentative energy metabolism genes. The presence of fermentative energy production in $N_2$-fixing cells is further supported by the significantly higher activities of all three key fermentative enzymes in $N_2$-fixing cells than in N-grown cells (Supplementary Fig. 8b). Furthermore, based on the activities of three different fermentative enzymes, it is reasonable to conclude the existence of a mixed acid type fermentation, a typical fermentation mode in bacteria[34,35]. In conclusion, in $N_2$-fixing Cyanothece sp. cells under dark and anaerobic conditions, it is highly probable that aerobic respiration plays a role only in the initial period of the dark phase, probably to power the synthesis of nitrogenase and partial $N_2$ fixation and to create an anoxic interior, while mixed-acid fermentation, which requires further validation, coupled with $Na^+$ energetics, functions to power $N_2$ fixation in the rest of the time.

## Discussion

Contrary to the traditional view that coastal $N_2$ fixation is sensitive to high levels of NaCl[4,5], this study reports that $N_2$ fixation by a coastal unicellular $N_2$-fixer is exclusively dependent on the presence of $Na^+$. In the absence of $Na^+$, the cells failed to fix $N_2$, were N-starved and showed a stringent response (or chlorosis), despite producing substantially higher levels of active nitrogenase. Further experiments show that the presence of $Na^+$ is a prerequisite for the activation of $Na^+$ energetics for the generation of ATP, which is subsequently used for the energy-intensive $N_2$ fixation. Thus, this study demonstrates the exclusive and essential role of $Na^+$ energetics in powering $N_2$ fixation in unicellular cyanobacteria.

Conventionally, $H^+$ energetics are thought to be chemically more advantageous than $Na^+$ energetics[26]. So why should Cyanothece sp. cells capable of $H^+$ energetics (Fig. 1a and Fig. 6b) perform $Na^+$ energetics? In Cyanothece sp., as in many unicellular $N_2$-fixing cyanobacteria and some multicellular species, $N_2$ fixation occurs in the dark[16]. The necessary ATP is believed to be produced by $H^+$ energetics, coupled with the aerobic respiratory breakdown of endogenous glycogen granules, which not only energizes this process but also consumes intracellular $O_2$, creating an anoxic interior (so-called respiratory protection)[15,36]. Although an anoxic environment is required for nitrogenase to function, this process is very energy-intensive. Therefore, if this energy were produced solely by aerobic respiration of the glycogen granules, the cells would have to increase their respiration and hence their respiratory $O_2$ uptake. Simultaneously, the $O_2$ level in the cell should be kept low enough to avoid inactivation of the nitrogenase. Consequently, the demand for more $O_2$ for increased aerobic respiratory activity is in conflict with the demand for an anaerobic interior. These two processes, which should work in tandem, seem to be incompatible. Therefore, it is reasonable to assume that under $N_2$ fixation conditions (dark and anoxic), $H^+$ energetics (coupled to aerobic respiration of glycogen) is not favoured and instead $Na^+$ energetics (coupled to anaerobic fermentation of glycogen), which can function independently of aerobic respiration[25], appears to be an alternative. Indeed, by integrating published data[17,30–33] and our results (Supplementary Fig. 8), it is reasonable to infer that although the cells were grown aerobically, mixed-acid fermentation rather than aerobic respiration mainly powers $N_2$ fixation, whereas active aerobic respiration only functions at the beginning of the dark phase.

Furthermore, under dark and anaerobic conditions, $Na^+$ energetics is not inferior to $H^+$ energetics from an energetic point of view. Only when there is a direct mechanistic link between redox reactions (oxidation of external resources) or photosynthesis in photosynthetic microbes and the translocation of $H^+$ across the membrane is $H^+$ energetics chemically more advantageous than $Na^+$ energetics[26]. Thus,

an energetic advantage only exists when high potential electron acceptors, such as $O_2$, are available. This is probably the reason why $H^+$ is mainly used as a coupling ion in the aerobic microbial world. In Cyanothece sp. under conditions where N is abundant, $O_2$ does not interfere with cellular metabolism during the day or night, allowing a more efficient use of $H^+$ energetics. However, when N is depleted, an obligate anaerobic environment is required for oxygen-sensitive $N_2$ fixation. In this case, $Na^+$ energetics coupled to fermentation are engaged, and organic intermediates of the glycolysis[37], such as pyruvate, serve as electron acceptors.

From an energetic perspective, the advantage of using both $H^+$ and $Na^+$ energetics in Cyanothece sp. can be rationalised by considering the energy challenges faced by cells in their complex natural habitats. To meet the energy demands of dynamic coastal seas, in particular the energy-intensive anaerobic $N_2$ fixation, organisms require a versatile energy metabolism, i.e., $H^+$ energetics coupled to photosynthesis and aerobic respiration during the day, and $Na^+$ energetics for anaerobic $N_2$ fixation in the dark. This metabolic flexibility likely contributes to the evolutionary resilience of cyanobacteria in a wide range of environments.

In the study of $Na^+$ energetics in cyanobacterial $N_2$ fixation, although three species of filamentous $N_2$-fixing cyanobacteria were reported to be dependent on NaCl for $N_2$ fixation about forty years ago[38], it was then concluded that inhibition of combined phosphorus (P) uptake in the absence of $Na^+$ was responsible for the loss of nitrogenase activity. However, this cannot explain the $Na^+$ dependence of $N_2$ fixation in Cyanothece sp. as we show here that P uptake is independent of the presence of $Na^+$ (Fig. 1a, Fig. 3c). Recently, although not directly related to $N_2$ fixation, a study using Synechocystis sp. PCC 6803 reported that N-starved cells engage in $Na^+$ energetics for ATP synthesis to maintain viability and to awaken from dormancy[20,39]. These reports suggest that $Na^+$ energetics may be more widespread in cyanobacteria than commonly thought. Indeed, $Na^+$ energetics is considered to be the most ancient form of bioenergetics, and genomic analyses suggest that it is still prevalent throughout the tree of life today[26,40].

Overall, the reported $Na^+$ energetics provides experimental evidence for their existence in microbial phototrophs in general, and the previously undescribed evidence for their role in cyanobacterial $N_2$ fixation in particular, providing a different perspective to understand the low abundance of cyanobacterial $N_2$-fixers in N-limited coastal waters. In nature, low coastal salinities caused by freshwater input (fluviatile and terrestrial freshwater)[41,42] and other environmental factors (such as temperature variations, the presence of predators) may together explain the low abundance of Cyanothece sp. Furthermore, the results presented also suggest that sodium energetics may be more widespread than expected, given its probable evolutionary origin in LUCA, and that its metabolic importance in other organisms may still be underestimated.

## Methods

### Strain and culture conditions

Cyanothece sp. ATCC 51142 was obtained from the ATCC culture collection. Stock cells were grown photoautotrophically at continuous light with a light intensity of 30 μmol $m^{-2}$ $s^{-1}$ in artificial seawater medium ASP2[9] at 30 °C. All experiments were carried out in a light-dark cycle of 12 h/12 h to accommodate the temporal segregation of photosynthesis and $N_2$ fixation, unless stated otherwise. The cells were cultivated without forced aeration, with the sole source of $CO_2$ being diffusion from the atmosphere into the flasks.

### N-dependent effects of NaCl on population growth

To evaluate the effect of NaCl on the growth and $N_2$ fixation activity of Cyanothece sp., we performed growth experiments in which we grew cells in artificial seawater medium ASP2 and artificial freshwater

medium BG11[43], supplemented with or without N, respectively. In total, we had four media, i.e., ASP2, ASP2 without N (ASP2-N), BG11 and BG11 without N (BG11$_0$). Prior to growth experiments, exponentially growing stock cells were harvested by centrifugation, washed twice to remove NaCl and resuspended in these four sterile media. The resuspended cells were subsequently inoculated into tissue culture flasks (Thermo Scientific, USA) containing 10 mL of the respective medium with a starting density of approximately $4 \times 10^6$ cells/mL. Cultures were grown aerobically at a light intensity of 30 μmol m$^{-2}$ s$^{-1}$ in a light-dark cycle of 12 h/12 h on a shaker at 150 rpm (stroke: 26 mm, MQD-S3R, Shanghai Minquan Instruments Co., Ltd) at 30 °C (unless otherwise specified, the same culture parameters were used in all experiments). Growth experiments lasted for nine days, and we recorded cell density daily using a cell counting chamber (Neubauer haemocytometry) under a light microscope (Axio observer Z1, Carl Zeiss, Germany). It should be noted that the counting of cells does not take into account the potential for variation in cell size across different experimental conditions.

To investigate the effects of NaCl deficiency on growth in the absence of N, cells previously grown in BG11$_0$ for seven days were subcultured in BG11$_0$ supplemented with either NaCl (18 g/L) or NaNO$_3$ (1.5 g/L or 17.6 mM). BG11$_0$ without extra additives was used as a blank control. Each culture was inoculated with a starting density of ~ $5 \times 10^5$ cells/mL. Growth experiments were conducted in triplicate for seven days. Cell density was quantified as described above.

To investigate the effect of NaCl on the growth of *Cyanothece* sp. in the presence of N, we conducted experiments with three replicates (with an initial density of approximately $4 \times 10^6$ cells/mL) for each of the fourteen NaCl concentrations (0, 2, 4, 9, 18, 20, 22, 24, 26, 28, 30, 32, 34, 36 g/L) in BG11 medium. The experiments lasted seven days, after which cell density was quantified and the increase in cell number was determined. Similar experiments were conducted using BG11$_0$ medium instead of BG11 to evaluate the impact of NaCl on the growth of *Cyanothece* sp. in the absence of N. These experiments lasted seven days, during which cell density and population growth were recorded and calculated. Population growth (the increase in cell number) was determined by subtracting the cell density on day 0 from the cell density on day 7.

To investigate whether the NaCl requirement for N$_2$ fixation depends on cellular growth stage, triplicate populations of cells (with an initial density of approximately $4 \times 10^6$ cells/mL) from exponential phase (7 days after inoculation) or stationary phase (20 days after inoculation) were grown in BG11$_0$ medium with or without NaCl addition. This experiment lasted for seven days, during which cell density and population growth were recorded and calculated as described above.

### Measurement of N$_2$ fixation activity

N$_2$ fixation activity of cells cultured in BG11$_0$ with seven NaCl concentrations (0, 2, 4, 9, 18, 20, 22 g/L) was measured by a modified acetylene reduction method. Cultures were grown for five days under the conditions described above and cell density was quantified on the last day. Assays were conducted in Vacutainer tubes containing 2 mL of the corresponding cell samples and 15% acetylene in the gas phase. Each sample was incubated for 6 hours under the same conditions of light, temperature, and shaking as in the growth experiments. 0.2 mL of triplicate samples were analysed for ethylene concentration by high-performance liquid chromatography (HPLC). The rates of nitrogenase activity are reported in nanomoles of C$_2$H$_4$ produced per $10^8$ cells per hour.

### Cellular chlorophyll, total protein, dry weight

To quantify cellular chlorophyll, 1 mL of cells from 7-day-old *Cyanothece* sp. cell cultures grown in either BG11$_0$ or BG11$_0$ with 18 g/L NaCl medium (hereafter BG11$_0$ (NaCl)), were centrifuged and then extracted twice with 5 mL of 80% aqueous acetone. The extracts were pooled and the spectra of this extract and a sample of whole cells were measured using a DW2000 spectrophotometer (Olis, GA, USA), both against 80% acetone or BG11 media as reference. Chlorophyll *a* and *b* levels from the acetone extracts were calculated as outlined in a previous study[44]. Specifically, the concentration of chlorophyll was determined using the equations:

$$\text{Chl } a(\mu g/mL) = 12.25A_{663} - 2.79A_{647}; \qquad (1)$$

$$\text{Chl } b(\mu g/mL) = 21.5A_{647} - 5.1A_{663}; \qquad (2)$$

$$\text{Chlorophyll} = \text{Chl } a + \text{Chl } b; \qquad (3)$$

where $A_n$ is the absorbance spectrophotometrically measured at the specific wavelength of 663 and 647.

Total protein content was determined using a Bradford Protein Assay kit (Solarbio, China) following the manufacturer's instructions. Cells grown in either BG11$_0$ or BG11$_0$ (NaCl) for 7 days were collected and cell density was recorded. Then, 2 mL of culture from each treatment was used to extract total protein using a Plant Protein Extraction Kit (Solarbio, China). Specifically, 2 mL of sample was centrifuged and the supernatant was discarded. Glass beads were used to grind the samples and thoroughly break the cell wall under liquid nitrogen. 1 mL of lysis solution was added at 4 °C for 20 minutes, samples were shaken every 5 minutes, followed by centrifugation at 4 °C at 20000 × g for 30 minutes. The supernatant was harvested for quantification. All experiments were done in triplicate.

To quantify cell dry weight, 10 mL of cells grown in either BG11$_0$ or BG11$_0$ (NaCl) were concentrated in pre-weighed Eppendorf tubes and cell density was quantified. Then, each treatment was centrifuged at 20000 × *g* for 5 minutes. The resulting pellet was washed twice with ddH$_2$O and the supernatant was discarded. The tubes were dried in an oven at 65 °C for 24 hours and the total dry weight (cells + tubes) was subsequently determined gravimetrically. Cellular dry weight was calculated with the following equation: Cellular dry weight = (Total dry weight – Weight of tube) / cell number.

### Metal chlorides, NH$_4$Cl, Glutamine addition

To explore the mechanisms of growth inhibition in the absence of NaCl under diazotrophy, we grew cells in BG11$_0$ and provided them with additional substances to determine which part of the metabolism is affected by NaCl deficiency. To this end, we assessed whether NaCl could be replaced by other metal chlorides by adding an equivalent amount (310 mM) of KCl, LiCl, MgCl$_2$ and CaCl$_2$. Moreover, to investigate whether low concentrations of LiCl could stimulate population growth under diazotrophy, a gradient of LiCl (0, 5, 10, 20, 50, 100, 200, 300 mM) was added to cells grown in BG11$_0$ medium. Furthermore, to investigate whether N$_2$ fixation products can lead to population growth, we supplemented the BG11$_0$ medium with additional NH$_4$Cl (2 mM) or glutamine (2 mM, Aladdin, China). The concentration (2 mM) of NH$_4$Cl or glutamine was chosen based on the result of a pre-experiment, which was neither too low to support population growth nor too high to potentially inhibit population growth. We conducted 7-day-long growth experiments using three replicates (with an initial density of approximately $4 \times 10^6$ cells/mL) in these experimental set-ups, cell densities were recorded and the increases in cell number were calculated.

### Sparging experiment

For anaerobic incubation, cells with a starting density of approximately $4 \times 10^6$ cells/mL were transferred into tissue culture flasks with either BG11$_0$ or BG11$_0$ supplemented with 18 g/L NaCl. A total of four

treatments were included, i.e., sparging (BG11$_0$, BG11$_0$ (NaCl)) and no sparging (BG11$_0$, BG11$_0$ (NaCl)). For the two sparging treatments, tissue culture flasks were flushed with N$_2$ for 10 minutes and then sealed with parafilm to prevent diffusive influx of O$_2$. Triplicate cultures were cultivated for seven days under the conditions above. Afterwards, the cell density of each treatment was quantified.

### Quantification of nitrogenase content and estimation of nitrogenase activity

The enzyme activities and contents of nitrogenase of *Cyanothece* sp. collected in the dark period were measured using commercial nitrogenase activity ELISA (enzyme-linked immunosorbent assay) kit (RF12981, Shanghai Ruifan Biological Technology Co., Ltd, Shanghai, China) and nitrogenase content ELISA kit (MM-246202, Jiangsu Meimian Industrial Co., Ltd, Yancheng, China), respectively. The NaNO$_3$ treatment is used to assess the accuracy of the ELISA approach, as cells should not synthesise nitrogenase when N is replete. Enzymes were extracted from cells grown in the respective media (approximately $1 \times 10^8$ cells) in an Eppendorf tube containing 1 mL of ice-cold PBS buffer (pH = 7.4). The homogenate was centrifuged at $10000 \times g$ for 5 minutes at 4 °C and the resulting supernatant was used for further assays. 50 μL of standard, sample and control (nothing added) were added to a well of the nitrogenase antibody pre-coated 96-well plate. The HRP (horseradish peroxidase)-conjugated reagent was then added and the whole plate was incubated at 37 °C for 30 minutes. Detection reagents A (50 μL) and B (100 μL) were added sequentially to each well, incubated for 30 minutes at 37 °C and washed with wash buffer for 5 times. 90 μL of TMB (tetramethylbenzidine) substrate was added and incubated for 15 minutes at 37 °C, after which 50 μL of stop solution was added. Finally, the OD (optical density) at 450 nm was measured and the nitrogenase concentration and activity were calculated from the standard curve. For each condition, six biological replicates were measured.

### Transcriptomics

20 mL of cells (seven-day-old since inoculation) grown in either BG11$_0$ or BG11$_0$ with NaCl (18 g/L) were harvested in the dark phase and then centrifuged at $20000 \times g$ for 10 minutes at 4 °C to remove the supernatant. Cell pellets were resuspended in ddH$_2$O and centrifuged again. The supernatants were discarded to remove residual medium. This procedure was repeated twice. The collected cell pellets were transferred to RNase-free cryogenic vials and fast frozen in liquid nitrogen for 30 minutes until extraction. In total, we had both cell types in triplicate.

Total RNA was extracted using Plant RNA Purification Reagent for plant tissue (Invitrogen, USA), and genomic DNA was removed using DNase I (TaKara, Dalian). Only high quality RNA samples (OD260/280 = 1.8 - 2.2; OD260/230 ≥ 2.0; RIN ≥ 6.5; 28S:18S ≥ 1.0; and > 1 μg) were used for sequencing library construction. Total RNA samples were sent to Shanghai Majorbio Bio-pharm Technology Co., Ltd. (Shanghai, China). Paired-end sequencing was performed on the Illumina sequencing platform (San Diego, CA, USA) according to the manufacturer's instructions (Illumina).

Data were analysed using the online platform Majorbio Cloud [www.majorbio.com][45]. Raw paired-end reads were trimmed and quality controlled for adaptor contamination using SeqPrep [https://github.com/jstjohn/SeqPrep] and Sickle [https://github.com/najoshi/sickle] with default parameters. Clean reads were then aligned to the *Cyanothece* sp. ATCC 51142 genome using HISAT2 software [http://ccb.jhu.edu/software/hisat2/index.shtml]. Mapped reads from each sample were assembled using StringTie [https://ccb.jhu.edu/software/stringtie/index.shtml?%20t=example] in a reference-based approach.

To identify the differentially expressed genes (DEGs) between the two different treatments, the expression level of each transcript was calculated using the transcripts per million reads (TPM) method. RSEM was used to quantify gene abundance. Essentially, differential expression analysis was performed using DESeq2 with a Q-value ≤ 0.05, and significant DEGs were confirmed only if |log2FC| > 1 and a Q-value ≤ 0.05. Finally, KEGG pathway analyses were performed using Goatools [https://github.com/tanghaibao/Goatools] and KOBAS, respectively. TPM was used for data visualisation. Heatmaps were generated using ComplexHeatmap v2.8.0. Heatmap Z-scores were calculated for each gene by subtracting the gene expression from the row mean and then dividing by the row standard deviation. All other plots were generated using ggplot2 v3.3.5.

### Glycogen quantification

Glycogen content was quantified using a hexokinase activity assay kit (BC0745, Solarbio, Beijing, China). Triplicate populations were cultured in BG11$_0$ or BG11$_0$ (NaCl) in a 12 h/12 h light/dark cycle for 3 days. 10 mL samples of each treatment were collected at the end of the light phase and at the end of the dark phase, respectively. The collected cells were pelleted and washed with distilled water. 30% (weight/volume) KOH was added to lyse the cells, followed by incubation at 95 °C for 90 minutes. Glycogen was precipitated by addition of cold absolute ethanol for 2 hours on ice and collected by centrifugation (15000 g for 10 minutes). The pellets were washed with absolute ethanol and dried at 60 °C. 300 μL of 100 mM sodium acetate buffer (pH 4.75) was added to resuspend the pellets, and 25 μL of the resuspended samples were used to measure the background glucose level. The remaining glycogen samples were digested with amyloglucosidase at 55 °C for 25 minutes. Then 25 μl of sample was mixed with the assay reagent in a light-proof microtitre plate followed by incubation at room temperature (15 minutes). NADPH levels were measured at 340 nm using a DW2000 spectrophotometer (Olis, GA, USA) and the corresponding glycogen levels were calculated.

### ATP addition experiment

To examine whether the inhibition of nitrogenase activity was caused by the lack of ATP, additional ATP (0.1 mM, Solarbo, China) was added to BG11$_0$. As the commercially available aqueous ATP solution has a concentration of 10 mM and the volume of BG11$_0$ medium used in the experiment is 10 mL, a final concentration of 0.1 mM ATP was chosen to alleviate the dilution of the culture medium by the addition of the ATP mother solution and to ensure a sufficiently high ATP concentration to support population growth. Here, 100 μL of ATP mother solution was added to 9900 μL of each culture medium. We tested the growth of three replicates (with an initial density of approximately $4 \times 10^6$ cells/mL) in the modified BG11$_0$ media. After seven days, the cell density was quantified and the increase in cell number was calculated.

To determine the actual role of ATP in activating population growth, cells (approximately $4 \times 10^6$ cells/mL) were cultured in BG11$_0$ modified with NaCl (310 mM), glutamine (0.1 mM), NaNO$_3$ (0.1 mM) or ATP (0.1 mM). To ensure that all treatments had the same concentration of additives, 0.1 mM glutamine and NaNO$_3$ were used in accordance with the concentration of ATP, as outlined above. Each treatment had six replicates. Cells were harvested in the dark period on day 3 for quantification of nitrogenase content by ELISA according to the manufacturer's protocol (Shanghai Ruifan Biological Technology Co., Ltd, Shanghai, China). If the role of ATP is to provide chemical energy, then nitrogenase biosynthesis would be expected; if ATP acts as an N resource, then nitrogenase synthesis is unnecessary. To further examine whether supplemented ATP functions as N or energy source, 0.1 mM (as outlined above) of ADP (Macklin, China) and AMP (Macklin, China) and ATP were added to the cells (a starting density of approximately $4 \times 10^6$ cells/mL) grown in BG11$_0$. Each treatment had three replicates. After seven days, the increase in cell number was calculated and contrasted.

## ATP and ADP quantification

To test whether $Na^+$ directly mediates the intracellular ATP content, we quantified the cellular ATP and ADP content of cells in different NaCl gradients. Cells with a starting density of $4 \times 10^6$ cells/mL were first cultured in $BG11_0$ for seven days. Since $N_2$ fixation occurs in the dark, NaCl (18 g/L) was added to these non-growing cultures in the dark phase on day 7. Cells were sampled at four time points after the NaCl addition (0 minutes, 10 minutes, 30 minutes, 60 minutes). 1 mL of cells (enumerated under the microscope) were snap-frozen in liquid nitrogen for 20 minutes and stored at $-80\,°C$ until ATP and ADP quantification. For ATP extraction, samples were subjected to three consecutive cycles of boiling at $100\,°C$ and freezing in liquid nitrogen, followed by centrifugation at $4\,°C$ at $20,000 \times g$ for 2 minutes. The resulting ATP supernatant was quantified using an established protocol of the ATP content assay kit (AKOP004M, Boxbio, Beijing, China). $200\,\mu L$ of ATP supernatant and a standard sample (0.625 µmol/mL ATP in $ddH_2O$) were incubated with a working buffer containing hexokinase, glucose, glucose dehydrogenase, and NADP, respectively. The absorbance of the synthesised NADPH was measured at 340 nm using a spectrophotometer at two time points: 10 s and 190 s after incubation. The ATP standard curve was generated and used to calculate the ATP content in the collected samples. For every condition, three biological replicates were measured.

For ADP quantification, cellular ATP and ADP were first extracted as described above, and ADP was subsequently quantified using an ADP Assay Kit (MAK133, Sigma-Aldrich, Missouri, USA) according to the manufacturer's protocol. Briefly, $10\,\mu L$ of the samples together with $10\,\mu L$ of standard (24 µM ADP) were mixed with $90\,\mu L$ of a working mix containing reaction buffer, luciferin and firefly luciferase, followed by incubation for 10 minutes at room temperature in a 96-well plate. The luminescence was quantified with a spectrophotometer (Olis, GA, USA) to determine the read luminescence for ATP ($RLU_{ATP}$). $5\,\mu L$ ADP reagent was then added to each well immediately after reading ($RLU_{ATP}$) and incubated for 2 minutes at room temperature to determine the read luminescence for ADP ($RLU_{ADP}$). The ADP standard curve was generated. The luminescence of ADP present in the samples was calculated ($RLU_{ADP} - RLU_{ATP}$), and the corresponding ADP content was determined using the standard curve. Three biological replicates were measured for each condition.

## Exposure to monensin, DTHB, EIPA and DCCD

The following assays were performed to understand the role of $Na^+$ or $H^+$-coupled ATP synthesis in driving ATP generation in the presence or absence of $Na^+$ under diazotrophy. The $Na^+$-specific ionophore monensin (Solarbo, Beijing, China), an inhibitor of $Na^+$ channels and the $Na^+/H^+$ antiporter ethyl-isopropyl amiloride (EIPA, GLPBIO, USA) and the protonophore 3,5-di-tert-butyl4-hydroxybenzaldehyde (DTHB, Thermo Fisher Scientific, USA) were used to test which pathway drives ATP synthesis. A typical $F_0F_1$ ATP synthase inhibitor, dicyclohexylcarbodiimide (DCCD, Macklin, China), was used to test whether the $H^+$-motive and $Na^+$-motive machinery share a common ATP synthase. Firstly, to quantify changes in cellular ATP levels and ATP/ADP ratios, monensin (14 µM) or EIPA (100 µM) was added to $N_2$-fixing cells (with a starting density of approximately $4 \times 10^6$ cells/mL) grown in $BG11_0$ with 18 g/L NaCl in the dark phase on day 7. 1 mL of cells were sampled at four time points after the addition of monensin or EIPA (0 minutes, 10 minutes, 30 minutes, 60 minutes). Cellular ATP and ADP levels were quantified as described above. Secondly, to understand the energetics of cells under different conditions, we tested the growth performance of cells in different media with the addition of monensin, DTHB, EIPA or DCCD. We used four media, including BG11, BG11 with NaCl (18 g/L), $BG11_0$, $BG11_0$ with NaCl (18 g/L), and cells (with a starting density of approximately $4 \times 10^6$ cells/mL) were cultured in each treatment with either monensin (14 µM), DTHB (200 µM), EIPA (100 µM), DCCD

(100 µM) or no additive (as blank control). After seven days of growth, the increase in cell number was recorded and contrasted. To test whether monensin interferes with the performance of nitrogenase, cells (with a starting density of approximately $2.5 \times 10^6$ cells/mL) were cultured in $BG11_0$, $BG11_0$ (0.1 mM ATP) and $BG11_0$ (0.1 mM ATP & 14 µM monensin), respectively. After three days of growth, the increase in cell number was recorded and compared.

## Estimation of the activity of fermentation-related enzymes

To determine the activities of fermentation-related enzymes, cells (approximately $4 \times 10^6$ cells/mL) were cultured in BG11 with NaCl (18 g/L) and in $BG11_0$ with NaCl (18 g/L), respectively. Each treatment had nine replicates. The experiment was carried out in a 12 h/12 h light-dark cycle, and cells were harvested twice on day 7 at midnight to estimate the activities of three fermentation-related enzymes, i.e., acetate kinase, alcohol dehydrogenase, and lactate dehydrogenase, using ELISA, according to the manufacturer's protocol (MM-1181O2, MM-91569O2, MM-2023O2, Jiangsu Meimian Industrial Co., Ltd, Yancheng, China).

## Statistical analysis

Physiological parameters were analysed using two-tailed Welch's $t$ test. Growth data, nitrogenase activity data (acetylene reduction method) and ELISA data were analysed by one-way ANOVA. Monensin, DTHB, EIPA and DCCD data were analysed by two-way ANOVA. Significance: ns (no significance), *($p < 0.05$), **($p < 0.01$), ***($p < 0.001$), ****($p < 0.0001$). All statistical analyses were performed in R v4.1.1[46].

## Reporting summary

Further information on research design is available in the Nature Portfolio Reporting Summary linked to this article.

# Data availability

Raw RNAseq reads for the analysis of differential gene expression have been submitted to NCBI's SRA database (http://www.ncbi.nlm.nih.gov) under BioProject PRJNA1014498. The KEGG database [https://www.kegg.jp/] was used for functional enrichment analyses. The data generated in this study are provided in the Source Data file. Source data are provided with this paper.

# Code availability

All scripts for data visualisation are available at https://github.com/SiTANG1990/N2-fixation-coastal-unicellular-cyanobacteria.git [https://doi.org/10.5281/zenodo.13923109[47]].

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

## Acknowledgements

We thank Yuelu Jiang and Peter Deines for their helpful comments on this paper. This work was supported by the S&T Projects of Shenzhen Science and Technology Innovation Committee (KCXFZ20211020163557022) to ZH.C. K.H. gratefully acknowledges the financial support of the John Templeton Foundation (#62220). The opinions expressed in this paper are those of the authors and not those of the John Templeton Foundation.

## Author contributions

S.T., K.H., and ZH.C. conceptualised and designed the experiments. S.T., X.Y.C., YQ.L., L.L., Q.Y., D.L., and JM.Z. (Jianming Zhu) performed the

experiments. S.T., K.H., J.Z. (Jin Zhou), and YY.J. analysed and visualised the data. All authors interpreted the results and wrote the manuscript.

## Competing interests

The authors declare no competing interests.
