## [Transparent Peer Review file · Nature Communications]

A unicellular cyanobacterium relies on sodium energetics to fix N₂

Corresponding Author: Professor Zhonghua Cai

Version 0:

Reviewer comments:

Reviewer #1

(Remarks to the Author)

The paper entitled "Unicellular cyanobacteria rely on sodium energetics to fix N₂" proposed new insight into mechanism for providing energy to a nitrogen fixation in unicellular type marine cyanobacterium. The reviewer listed comments and questions for the contents. Please address them.

Major

1. About cause of ATP production for nitrogen fixation: The authors insisted that Na⁺ energetics is the cause of ATP production under N depletion/salinity condition. But I think ATP from glycolysis cannot be excluded. The authors discussed this in lines 498-512, but how about confirming by experiment? For example, what result is obtained by putting glucose as C source into BG11-0 medium supplemented with photosynthesis inhibitor? Please consider.
2. Fig. S7, lines 437-445: If I am not mistaken, the panel A's NaCl and the panel B's N- are same condition (BG11-0 + NaCl condition). Why ACKase activity was very high although transcript of ark was not detected? Please explain.
3. About effects of abiotic factors: In lines 447-470. The authors showed the abiotic factors in the ocean, but I feel experimental data is needed for connecting them to cyanobacterial population. In addition to data in Fig S8d, data mimicking conditions in nature with NaCl and temperature will strongly support the hypothesis.

Minor

1. Line 156: Are there any other pho genes in this cyanobacterium?
2. Line 264: 'and' should be 'or'?
3. Lines 342-350: The authors proposed two candidates of Na⁺ gradient generator. I think a number of cation/proton transporters are in Cyanobacteria. Are there any other candidates in 1909 up-regulated DEGs under salinity condition?
4. Line 397: What is 'evidence'?
5. Fig. 5, panel a: 'Percentage of species' is correct?

Reviewer #2

(Remarks to the Author)

Tang et al.

The authors extensively investigate the role of Na⁺ energetics in cyanobacterial metabolism, specifically in nitrogen fixation. Many strains of cyanobacteria are capable of fixing atmospheric nitrogen to a biologically accessible form. This ability therefore significantly contributes to the oceanic biological nitrogen cycle. Unicellular cyanobacteria have recently been recognized for contributing to nitrogen fixation in marine environments, a function previously thought to be performed exclusively by filamentous cyanobacteria.

Of the unicellular nitrogen-fixing cyanobacteria, Cyanobacteria constitutes an important genus. One Cyanobacteria strain,

Cyanothece sp. ATCC 51142 (Cyanothece 51142) has been comprehensively studied at the genomic, transcriptomic, proteomic, including physiological levels. Cyanothece 51142 performs oxygenic photosynthesis and nitrogen fixation, separating these two incompatible processes temporally within the same cell.

A strong impairment of nitrogen fixing capacity in heterocystous nitrogen-fixing cyanobacteria under increased salinity was well reported. In contrary, this study for the first time clarifies that nitrogen fixation is exclusively rely on the presence of Na⁺ in a unicellular nitrogen-fixing cyanobacterium Cyanothece 51142. The authors also demonstrate the essential role of Na⁺ energetics in powering nitrogen fixation in Cyanothece 51142. Na⁺ energetics operating by Na⁺ pumping proteins (Na⁺-ATPase) rather than proton pumping (H⁺-ATPase) fuel nitrogen fixation in this cyanobacterium. Further, Na⁺-ATPase is essential to provide ATP for nitrogen fixation, which is likely coupled to anaerobic rather than aerobic respiration. Together, these findings are very interesting and provide insights into the crucial role of Na⁺ energetics in nitrogen fixation. The findings also offer valuable insights into the molecular and cellular mechanisms by which Na⁺ energetics participate in nitrogen fixation, highlighting their physiological significance. The data are convincing, but a few parts may need additional experiments and a more thorough discussion.

Points to be clarified:

1. The authors indicate that the ATP increase relies on a sodium motive force, and this sodium energetics fuel nitrogen fixation in Cyanothece 51142. To corroborate the role of sodium in ATP synthesis during nitrogen fixation, in addition to testing by monensin (a sodium ionophore) and DTHB (a protonophore), the authors should test effect of an inhibitor of sodium channel and Na⁺/H⁺ antiport (eg. amiloride).
2. Sodium requirement for nitrogen fixation depends on the cellular growth stage or not?
3. The authors suggest that functioning Na⁺ energetics requires both a Na⁺ gradient consumer by ATP synthase and Na⁺ gradient generators by Na⁺ pumps. The authors should show at least transcriptomes of significantly upregulated or downregulated DEGs for relevant genes.
4. Growth inhibition in the absence of NaCl, equivalent amount (310 mM) of KCl, LiCl, MgCl₂ and CaCl₂ was tested. LiCl is a structural analog of sodium chloride but it is a toxic salt. Decreased cell numbers observing in Fig3A was possibly resulted from too high concentration of LiCl.

Reviewer #3

(Remarks to the Author)

The paper by Tang et al describes the sodium dependence of nitrogen fixation in the unicellular cyanobacterium Cyanothece.

The core facts are the following:

The initial observation was that Cyanothece requires elevated concentrations of sodium ions in order to be capable of nitrogen fixation. Then the authors set out to find the basis of this sodium requirement. Therefore they performed a series of physiological studies, where they show that at low sodium, nitrogenase are highly expressed, the nitrogenase enzyme is present but it fails to fix nitrogen. Therefore, the cells become severely nitrogen starved and show the typical symptoms of nitrogen chlorosis (what they mistakenly term "stringent response", reason see below). By adding ATP or with inhibitor experiments, they present data that the failure of N₂ fixation is probably because of energy deprivation in at low sodium concentrations. At this point these are mostly hard facts, but the rest is pure speculation: they compare in silico the ATPases of cyanobacteria and classify them in different selectivity towards proteins or sodium ions. Unfortunately, there is no experimental evidence that the ATPase from Cyanothece can really use sodium gradient. Most ATPases are localized at the thylakoid membrane, where a proton motive force is maintained. They propose that an enigmatic Rnf complex generates a sodium motive force in the night by a sort of anaerobic respiration, without clarifying what this anaerobic respiration may be. In the discussion, the authors confuse anaerobic respiration with fermentation. Anaerobic respiration requires an alternative electron acceptor instead of oxygen. What should this be? It is generally accepted that aerobic respiration is a mechanism to deplete intracellular oxygen levels and create an anoxic environment in the cells. This has been clearly established in heterocysts. Whether the cells perform aerobic respiration or anaerobic respiration or anaerobic fermentation is solely dependent on the external availability of oxygen, or the presence or absence of other terminal electron acceptors, but this is no "free choice" of the bacteria, as the author insinuates.

Further to this odd part of the manuscript comes the entire chaotic discussion about the seasonal fluctuations in salinity and temperature in coastal waters as an explanation for low N₂ fixing activities. When you look carefully at the data (provided in supplemental Fig 8), the contrary of what the authors claim seems to be the case: the lowest salinity is in April/May, whereas at the end of summer, in August, September, where the temperatures are highest, salinity is also high. So there is no antagonism between temperature and salinity. The entire explanation that growth supporting high temperatures coincide with low salinity, therefore impeding N₂ fixation is obsolete.

To me, the entire study seems to be preliminary and strongly biased by the belief of the authors in their sodium hypothesis. There are alternative explanations for the sodium dependence that have not been tested: as N₂ fixation occurs in the night, the cells need high amounts of glycogen to fulfill the energetic need of nocturnal N₂ fixation. If glycogen synthesis is dependent on increased sodium concentrations (for example because of sodium-dependent bicarbonate transporters), then the sodium requirement could be indirect. Measuring the glycogen levels under the different growth conditions during day-night cycles is absolutely required.

Another pivotal requirement is to determine the oxygen concentration or oxygen requirement for nocturnal N₂ fixation. If the hypothesis of an anaerobic Rnf-complex based energy generation system is correct, then N₂ fixation would be optimal under anaerobic conditions. The consumption of oxygen during nocturnal N₂ fixation needs also to be determined. Instead,

the authors can skip the entire speculative paragraph on sodium dependent ATPase (without showing corresponding data) or the confusing discussion on the ecological consequences.

A few other comments:

The description of experimental parameters is confusing. N and P are no nutrients, but are elements. The nutrients are N-compounds such as nitrate, ammonium.... or phosphorous. The concentrations of the nutrients should be indicated. It is completely unclear how ATP should be taken up by the cells. Under N-poor conditions, it will be used as a nitrogen source. As it was supplied at concentrations of 0.1 mM (according the methods), this cannot be compared to the 17 mM nitrate in standard BG11.

There are several mistakes and confusions in the text, just to mention some of them:

Further specific points

I. 48: many bloom formers in fresh water are non-diazotrophic (e.g. *Microcystis*).

I. 60: high salinity doesn't generally suppress N₂ fixation in filamentous cyanobacteria. For example, *Trichodesmium* is actively fixing N₂ in seawater.

I. 71-75: here you turn the paradox upside down without solving it

The I- 143- 150: Here the authors confuse stringent response with nitrogen-starvation induced chlorosis. In all microbiology textbooks, stringent response is presented as the specific response mediated by the alarmone ppGpp. In heterotrophic bacteria, amino acid starvation is the most important trigger for elevated ppGpp and thus of the stringent response. However, in cyanobacteria, this is completely different: Here ppGpp coincides with nocturnal dormancy: the lack of energy in the absence of photons causes elevated ppGpp, as clearly shown for *Synechococcus* (Hood, Higgins et al., PNAS 2016 and similar reports). By contrast, nitrogen chlorosis is induced by a complex network of stress-responsive transcription factors, such as NblS or NtcA, which ultimately initiate chlorosis. This has been thoroughly elaborated in *Synechocystis*, see several recent reviews on this topic.

L. 264. Here you need to mention the concentrations of Gln or nitrate.

L. 360: Sentence is unclear. Do you mean "except its role" instead of "but see its role"

L. 371: What is "Low+Medium/High"? Do you mean, "low, medium or high" ?

Generally:

In the first part of the paper, it should be clearly stated how the cells were grown in diurnal cycles, and that N₂ fixation only occurs during the dark period.

The role of cyanophycin as transient nitrogen storage compound has been completely neglected.

Concerning the transcripts that you have investigated:

In the databases, no *rnf* gene is annotated for *Cyanothece*. Provide the accession numbers of the genes that you mention in this study, otherwise, it is not clear what you really looked up. Some subunits of Rnf complexes show similarities with subunits of Photosystem I complex. What is the evidence that this gene you termed "*rnf*" is really part of an Rnf complex?

Reviewer #4

(Remarks to the Author)

This manuscript reports the important relationship between N₂ fixation and Na⁺ energetics in unicellular coastal cyanobacteria. This is an interesting study concerning cyanobacterial physiology and is of interest from the view of the global marine N-cycle. Although comprehensive studies have been conducted, this reviewer would advise the authors to consider the following points.

General comments

The following two points are important and should be considered by the authors first.

1. Growth phenotypes shown in Figure 1 clearly show the strong effect of NaCl on both the N₂ fixation and the growth of *Cyanothece*. Then the authors focused on the nitrogenase activity and the amounts of ATP in the cell which may affect the N₂ fixation function. To confirm if a limitation of ATP led to N₂ fixation, they supplemented ATP into the culture medium and observed the stimulation of the growth. This is a very interesting result. However, since nothing is known about ATP transporters or carriers on the cell membrane, the interpretation of this result is a major issue. If supplementation of ATP in the medium increases the ATP content in the cell, the authors must first ascertain whether this ATP effect is simply the uptake of ATP from the medium or some signaling effect of ATP on the cell.

In addition, the energy level in the cell is not determined solely by the amount of ATP in the cell even if the supplemented ATP is actively taken up by the cell. The balance between ATP and ADP is critical, and the authors must at least directly determine the amounts of adenylates in the cell.

2. The second half of the manuscript focuses on Na⁺ energetics and Na⁺-driven ATP synthases. This is another very important perspective.

Unfortunately, the authors only focus on the amino acid sequence of the c-ring, which has already been studied in databases (Refs. 19, 32). On the other hand, it has already been reported and biochemically proven that several ATP synthases from bacteria living in specific environments are Na⁺ gradient-driven (Dimroth et al.). However, this is still a very rare case, and so far, nothing is known about cyanobacterial ATP synthases. Because the information provided by this manuscript is so important to understanding the physiology of cyanobacteria living in a Na⁺ environment, a solid biochemical basis would be needed to discuss cell growth by Na⁺-driven ATP synthase in this manuscript.

Minor comments

1. Fig. 1a and b Since it is very difficult to set the same growth conditions, a logarithmic graph is usually used to show the growth curve.
2. Fig. 1c-e To find the most useful Na⁺ conditions, these graphs should be shown in scatterplots.
3. line 242. Information about the measurement of nitrogenase content in the cell is very limited. Though the authors described the method in the 'materials and methods' section (see lines 665-666), it is unknown how they can determine the amounts of the desired protein by commercial ELISA. At least, the purified recombinant protein must be necessary as a standard for this measurement.
4. line 278 and supplementary Fig. 4a: ADP and AMP must be also examined.
5. line 293: The authors should consider the possibility that Na⁺ alters the H⁺ gradients in the cell and indirectly affects the ATP synthesis reaction. Direct biochemical measurement is necessary to conclude.
6. line 744: How can they determine the amounts of ATP in the cytosol by this method? It should be described precisely.
8. Supplementary Fig. 1 Although the authors simply determined the chlorophyll content in the cells, spectral analyses might be more useful for studying changes in the various colored pigments in the cells.
7. lines 446-470: This reviewer wonders if a discussion of the relationship between observed salinity dynamics in the Texas Gulf Coast and cyanobacterial growth based on laboratory data is appropriate as a topic for a separate section in Results. The figures shown in Supplementary Figures 8a-c are based on published data, and only Figure 8d was generated from original data by the authors themselves. Therefore, interpretations relating to the two should be written in the discussion.

Version 1:

Reviewer comments:

Reviewer #1

(Remarks to the Author)

The authors have addressed the reviewer's concerns.
The manuscript was revised properly and ready for the publication.

Reviewer #2

(Remarks to the Author)

The authors have addressed all of my questions.

Reviewer #3

(Remarks to the Author)

The paper by Tang et al has considerably improved as compared to the previous version. However, there are still important issues that need clarification:

1. Following sentence needs correction:

„They up-regulated the expression of ppGpp (Supplementary Fig. 4b, gene location: NC_010546.1 (21413 - 22405), NCBI), which is the signalling molecule for the stringent response encoding gene10.

The relA gene (NC_010546.1) does not encode ppGpp, but the ppGpp synthase/hydrolase enzyme. These enzymes are usually under a strict metabolic control (for example induced by non-aminoacylated tRNA bound to the ribosome). Therefore, induced expression of relA is not a direct proof of elevated ppGpp levels. To measure ppGpp directly, this metabolite needs to be quantified. The induced expression of relA can therefore only be taken as an indication of increased ppGpp levels.

2. I do not understand the logic of the following:

„As shown in Fig. 2c, elevated transcript levels of all four P limitation biomarkers were observed in these non-N₂-fixing cells, indicating that the cells were P-starved, likely as an indirect consequence of N starvation. This up-regulated pattern of these genes makes biological sense: starving cells struggle to meet nutrient demands by expressing higher levels of nutrient synthases or transporters11“

The Pi assimilating genes are very specific for Pi limitation....It's very unlikely that N-limitation induces Pi assimilation genes (if I am wrong, provide a reference for such cross-talk) Moreover, growth arrested cells do not consume phosphorous since nucleic acid synthesis is also stopped. Therefore, the up-regulated pattern makes no immediate biological sense. There must be another reason for up-regulation of Pi assimilation genes.... perhaps the need to take up sodium is linked to Pi assimilation?

3. Following point is unclear:

„This result, coupled with the fact that cells can grow normally in BG11 (Fig. 1a), suggests that Na⁺ is not involved in bicarbonate uptake and glycogen synthesis“

It is not specified, how cell were cultivated and supplied with inorganic carbon. Cells depend on active (sodium-dependent) bicarbonate uptake only at ambient CO₂ concentrations. With higher CO₂ supply, CO₂ can enter cells by diffusion.

4. How the cells make use of extracellular ATP is still a mystery. If they would take it up, they would also use the nitrogen, as the purines are a rich source of combined N. What would happen to all the ADP/AMP from ATP turn-over, once the high energy gamma-phosphate of ATP has been consumed? According to the authors hypothesis, it cannot be re-charged by ATPase reaction....

5. Concerning anaerobic N₂-fixation:

„In fact, Cyanothecce sp. cells have been previously shown to fix N₂ anaerobically²⁷. Here, cells can generate energy in the absence of O₂, suggesting the existence of an anaerobic energy metabolism. In addition, the N₂ fixation activity of anaerobically grown cells was no worse than that of aerobically grown cells....“

I am afraid that the authors misunderstood Ref. 27, which they give as a reference for anaerobic diazotrophy... In this work, only the nitrogenase activity assay was done under anaerobic conditions (flushing cells with argon) but not cell growth: Anaerobic nitrogenase assay is a routine assay to check if the oxygen protection system of nitrogenase is ok. However, the cells have not been cultivated under anaerobic conditions. If the authors want to proof their hypothesis of an anaerobic fermentative metabolism that supports nitrogenase, then they need to show it experimentally..

Reviewer #4

(Remarks to the Author)

In the revised manuscript, the authors take the reviewers' comments seriously and have responded to them. However, the following points still require further consideration.

1. Na⁺-dependent ATP synthesis

The data presented by the authors certainly indicate that N₂ fixation is affected by Na⁺-energetics. However, as pointed out by other reviewers, it is undetermined whether Na⁺-dependent ATP synthesis is actually taking place. For this reason, placing Figs. 7b and c at the end of the Results seems misleading; perhaps they should be indicated in Discussion.

2. Experimental conditions

The ion concentrations used in this study seem to be deterministic everywhere (e.g., L211, L217, L237, L317, L318, L322). An explanation may be in order.

3. Notation of Cell Concentration

In this manuscript, cell volume is expressed in terms of cell number (L688). Since differences in experimental conditions (salt concentration) used at various parts may affect the size of cells, it may not be appropriate to use simply the number of cells for quantification.

4. L31, "Further sequence alignment analysis of the ion-binding site of the ATP

synthase": Since this is a description of the ion selectivity of the c subunit, the term ion-binding site is inappropriate.

Response to reviewers:

Reviewer #1:

The paper entitled "Unicellular cyanobacteria rely on sodium energetics to fix N₂" proposed new insight into mechanism for providing energy to a nitrogen fixation in unicellular type marine cyanobacterium. The reviewer listed comments and questions for the contents. Please address them.

Thank you for your detailed response and thoughtful comments! We have incorporated your comments throughout the manuscript. Please see our detailed responses below.

Major

1. About cause of ATP production for nitrogen fixation: The authors insisted that Na⁺ energetics is the cause of ATP production under N depletion/salinity condition. But I think ATP from glycolysis cannot be excluded. The authors discussed this in lines 498-512, but how about confirming by experiment? For example, what result is obtained by putting glucose as C source into BG11-0 medium supplemented with photosynthesis inhibitor? Please consider.

Firstly, we apologise for the confusion here as we should have clearly stated that ATP for N₂ fixation comes from glycolysis (lines 280-288). In fact, both H⁺ and Na⁺ energetics are coupled to glycolysis and describe two different approaches for how ATP is generated. In general, the energy for ATP synthesis comes from a transmembrane electrochemical H⁺ or Na⁺ gradient generated by glycolysis, which drives the rotation of the membranous F₀ motor component of the F₁F₀ ATP synthase (Ballmoos et al., 2009). Based on the type of coupling ion, either H⁺ or Na⁺, two types of energetics are defined, i.e., H⁺ energetics or Na⁺ energetics. Thus, Na⁺ energetics is coupled to, but not independent of, glycolysis. In the case of N₂ fixation by *Cyanothece* sp., it is well established that diazotrophic cells synthesise and store carbohydrates in the form of glycogen via photosynthesis in the light, then the cells use

stored glycogen via glycolysis to generate ATP to power N₂ fixation in the dark. So you are right that glycolysis is crucial to power N₂ fixation. The novelty of our findings is that we find that Na⁺ energetics, but not the traditional H⁺ energetics, is coupled to glycolysis to generate ATP for N₂ fixation. To gain further insight into the role of glycolysis in powering N₂ fixation, instead of applying a photosynthesis inhibitor and providing additional glucose, we quantified the changes in glycogen levels of N₂-fixing cells in a light/dark cycle. As can be seen in Fig. 4a, the N₂-fixing cells consumed a considerable amount of glycogen. Detailed changes can be found in lines 280-306.

2. Fig. S7, lines 437-445: If I am not mistaken, the panel A's NaCl and the panel B's N- are same condition (BG11-0 + NaCl condition). Why ACKase activity was very high although transcript of *ark* was not detected? Please explain.

Yes, you are correct that the cells from these two treatments were cultured under the same conditions (BG11₀+NaCl). In particular, as shown in Supplementary Fig. 8a, although the transcript of the gene *ack* from the figure appears to be zero, this gene actually has a transcript of approximately 0.543 TPM, which is comparatively lower but not absent. Furthermore, after a more careful examination of our transcriptomic data, we find that two *ack* genes are actually annotated at different locations in the genome. With different transcript levels for cells grown in BG11₀+NaCl, which are 0.543 (the data shown) and 31.72 (TPM) on average, respectively. We regret that we have overlooked this information and have now included the new information in the updated figure (Supplementary Fig. 7a). We believe that the new information of the transcript level of 31.72 (TPM) can explain the ACKase activities under N₂-fixing conditions. Detailed changes can be found in lines 493-514.

3. About effects of abiotic factors: In lines 447-470. The authors showed the abiotic factors in the ocean, but I feel experimental data is needed for connecting them to cyanobacterial population. In addition to data in Fig S8d, data mimicking conditions in nature with NaCl and temperature will strongly support the hypothesis.

Thank you for your kind suggestion. You are quite right that population abundance

data from mimicking conditions in nature would strongly support the hypothesis. However, since we do not know other important abiotic and biotic information, such as the concentrations of combined N and P, the presence or absence of predators, and more importantly, the *in situ* abundance of *Cyanothece* sp., it will be very difficult to mimic the real situation in nature. Based on the comments of the other reviewers, it seemed to detract from the main message of the manuscript, so we decided to remove this part from the manuscript.

Minor

1. Line 156: Are there any other *pho* genes in this cyanobacterium?

Based on our annotation analysis, another *phoD* gene (K01113, KEGG) encoding alkaline phosphatase D [EC:3.1.3.1] is identified. However, since *phoD* is not typically used as a bioindicator gene in published studies, we did not include it in our manuscript.

2. Line 264: 'and' should be 'or'?

Modified as suggested.

3. Lines 342-350: The authors proposed two candidates of Na⁺ gradient generator. I think a number of cation/proton transporters are in *Cyanothece*. Are there any other candidates in 1909 up-regulated DEGs under salinity condition?

Another gene encoding a K⁺/H⁺ antiporter was identified as a DEG between two treatments, but was irrelevant for Na⁺ transport. Transcriptomic analysis did not identify any other Na⁺ pumps.

4. Line 397: What is 'evidence'?

For this paragraph, we wanted to draw an overall conclusion, thus the "evidence" here refers to all results presented in previous contexts. To avoid confusion, we modified it to 'In summary, based on all evidence provided in this study,'. Detailed changes can be found in lines 561-566.

5. Fig. 5, panel a: 'Percentage of species' is correct?

The Y-axis of Fig.5a is the number of species under each situation/total number of species, so "percentage of species" is correct. The exact number of species under each situation is shown as the number on each bar of the figure. This is explained in lines 571-576.

References:

1. Von Ballmoos, C., Wiedenmann, A. & Dimroth, P. Essentials for ATP Synthesis by F₁ F₀ ATP Synthases. *Annu. Rev. Biochem.* 78, 649–672 (2009).

Reviewer #2:

Tang et al.

The authors extensively investigate the role of Na⁺ energetics in cyanobacterial metabolism, specifically in nitrogen fixation. Many strains of cyanobacteria are capable of fixing atmospheric nitrogen to a biologically accessible form. This ability therefore significantly contributes to the oceanic biological nitrogen cycle. Unicellular cyanobacteria have recently been recognised for contributing to nitrogen fixation in marine environments, a function previously thought to be performed exclusively by filamentous cyanobacteria. Of the unicellular nitrogen-fixing cyanobacteria, *Cyanothece* constitutes an important genus. One *Cyanothece* strain, *Cyanothece* sp. ATCC 51142 (*Cyanothece* 51142) has been comprehensively studied at the genomic, transcriptomic, proteomic, including physiological levels. *Cyanothece* 51142 performs oxygenic photosynthesis and nitrogen fixation, separating these two incompatible processes temporally within the same cell.

A strong impairment of nitrogen fixing capacity in heterocystous nitrogen-fixing cyanobacteria under increased salinity was well reported. In contrary, this study for the first time clarifies that nitrogen fixation is exclusively rely on the presence of Na⁺ in a unicellular nitrogen-fixing cyanobacterium *Cyanothece* 51142. The authors also demonstrate the essential role of Na⁺ energetics in powering nitrogen fixation in *Cyanothece* 51142. Na⁺ energetics operating by Na⁺ pumping proteins (Na⁺-ATPase) rather than proton pumping (H⁺-ATPase) fuel nitrogen fixation in this cyanobacterium. Further, Na⁺-ATPase is essential to provide ATP for nitrogen fixation, which is likely coupled to anaerobic rather than aerobic respiration. Together, these findings are very interesting and provide insights into the crucial role of Na⁺ energetics in nitrogen fixation. The findings also offer valuable insights into the molecular and cellular mechanisms by which Na⁺ energetics participate in nitrogen fixation, highlighting their physiological significance. The data are convincing, but a few parts may need additional experiments and a more thorough discussion.

Thank you very much for your enthusiasm for our work and your evaluation of the manuscript! We have modified the manuscript according to your suggestions.

Points to be clarified:

1. The authors indicate that the ATP increase relies on a sodium motive force, and this sodium energetics fuel nitrogen fixation in *Cyanothece* 51142. To corroborate the role of sodium in ATP synthesis during nitrogen fixation, in addition to testing by monensin (a sodium ionophore) and DTHB (a protonophore), the authors should test effect of an inhibitor of sodium channel and Na⁺/H⁺ antiport (eg. amiloride).

Thank you for your suggestion. We first tested the effect of amiloride (EIPA) on cellular ATP levels and the ATP/ADP ratio of N₂-fixing cells in the presence of Na⁺ (Fig. 4e, f), the results show that treatment with EIPA (100 μM) markedly suppressed ATP synthesis, resulting in lower ATP/ADP ratios (Fig. 5e, f), substantiating the key role of Na⁺ energetics in generating ATP for N₂ fixation. We then tested the effect of EIPA on cell growth in different media, and the results show that no growth was observed in any of the treatments exposed to EIPA (100 μM) (Fig. 6c), suggesting that EIPA inhibited both the Na⁺/H⁺ antiporter and other essential cellular functions through EIPA. The lower cellular ATP levels and ATP/ADP ratios observed following EIPA treatment (Fig. 5e, f) may be attributed not only to the inhibition of Na⁺-dependent ATP synthesis but also to the suppression of other key metabolic processes. Detailed changes can be found in lines 369-418.

2. Sodium requirement for nitrogen fixation depends on the cellular growth stage or not?

Based on the results presented, we believe that sodium is a general requirement for N₂ fixation, independent of the cellular growth stage. Specifically, both normally growing cells (Fig. 1a) and N-starved cells (Fig. 1b) require Na⁺ for N₂ fixation. As the cells used for experiments in this study are from the exponential phase (7 days after inoculation), in our revised manuscript, we also included cells from the stationary

phase (20 days after inoculation) to test whether these cells also require Na^+ for N_2 fixation. As shown in Supplementary Fig. 1, we find that cells from the stationary phase also require Na^+ for N_2 fixation. In particular, N_2 -fixing cells from the stationary phase grew significantly slower in the presence of NaCl than cells from the exponential phase cells. We think that this is because stationary phase cells require more time to recover growth and therefore have a longer lag time. Detailed results are shown in lines 103-107 and Supplementary Fig. 1.

3. The authors suggest that functioning Na^+ energetics requires both a Na^+ gradient consumer by ATP synthase and Na^+ gradient generators by Na^+ pumps. The authors should show at least transcriptomes of significantly upregulated or downregulated DEGs for relevant genes.

Sorry for the confusion. However, in the manuscript we have proposed two potential Na^+ pumps, i.e., Rnf and Na^+/H^+ antiporter, and we have shown their transcriptomic pattern in Supplementary Figure. 5, which is also stated in lines 347-350 of the original manuscript. However, upon closer examination of Rnf, we find that our current transcriptomic evidence cannot conclusively support that the model organism harbours a Na^+ -pumping Rnf complex, as some subunits of Rnf complexes show similarities to subunits of the photosystem I complex. As a consequence, we have removed Rnf from the new version of the manuscript. Regarding the Na^+/H^+ antiporter, we now provide more detailed evidence showing that the expression of all four Na^+/H^+ antiporter subunit encoding genes were significantly upregulated in non- N_2 -fixing cells grown in BG11₀ compared to N_2 -fixing cells grown in BG11₀ (NaCl), indicating that these cells struggled to meet their energy requirements by attempting to establish the transmembrane Na^+ gradient. Detailed results can be found in lines 446-460.

4. Growth inhibition in the absence of NaCl, equivalent amount (310 mM) of KCl, LiCl, MgCl₂ and CaCl₂ was tested. LiCl is a structural analog of sodium chloride but it is a toxic salt. Decreased cell numbers observing in Fig3A was possibly resulted from too high concentration of LiCl.

Thank you for this suggestion. We now also cultured *Cyanothece* sp. cells in BG11₀ medium with a LiCl gradient (0, 5, 10, 25, 50, 100, 200, 300 mM). As depicted in Supplementary Fig. 5, the results show that cells are still unable to grow at low LiCl concentrations (5, 10 mM), indicating that LiCl, unlike NaCl, cannot activate N₂ fixation. Detailed changes can be found in lines 214-223.

Reviewer #3:

The paper by Tang et al describes the sodium dependence of nitrogen fixation in the unicellular cyanobacterium *Cyanothece*.

The core facts are the following:

The initial observation was that *Cyanothece* requires elevated concentrations of sodium ions in order to be capable of nitrogen fixation. Then the authors set out to find the basis of this sodium requirement. Therefore they performed a series of physiological studies, where they show that at low sodium, nitrogenase are highly expressed, the nitrogenase enzyme is present but it fails to fix nitrogen. Therefore, the cells become severely nitrogen starved and show the typical symptoms of nitrogen chlorosis (what they mistakenly term “stringent response”, reason see below).

Thank you for your critical review, which has motivated us to clarify the exposition of our work. Our response to the “stringent response” issue can be found further below.

By adding ATP or with inhibitor experiments, they present data that the failure of N₂ fixation is probably because of energy deprivation in at low sodium concentrations. At this point these are mostly hard facts, but the rest is pure speculation: they compare in silico the ATPases of cyanobacteria and classify them in different selectivity towards proteins or sodium ions.

In bioenergetics, the transmembrane gradient of two ions, i.e., H⁺ and Na⁺, can drive ATP synthesis, also known as H⁺ energetics and Na⁺ energetics, respectively. Since the ability of an ATPase to use H⁺ and Na⁺ as coupling ions is determined by the c ring (AtpH) of the F₀ complex of the ATPase (Ballmoos et al., 2009 and Doello et al., 2021), we wanted to investigate the prevalence of Na⁺ energetics in the cyanobacterial phyla, i.e., whether they can use H⁺ energetics, Na⁺ energetics or both. To this end, we conducted the AtpH sequence alignment analysis. First, we did not classify the c ring of different cyanobacteria (121 species were compared in total) into different

selectivities towards protons or sodium ions, but towards protons or sodium ions. Compared to the hardly possible experimental evaluation of numerous cyanobacterial species one by one, the AtpH sequence alignment analysis, as also conducted by Doello et al., 2021, is a fast and efficient approach to explore the Na⁺ energetics capability. It allows us to study 121 cyanobacterial species and obtain a more complete picture of Na⁺ energetics in the cyanobacterial phyla. Therefore, we respectfully disagree with your assertion that the sequence alignment analysis is “pure speculation”. Instead, we think that these results are meaningful and contribute to the topic of cyanobacterial Na⁺ energetics. It seems to us that you think that we use the protein sequence similarity as the key evidence to support the Na⁺ energetics proposal, which is not the case. We apologise for the confusion we might have caused and have tried to clarify why we have done this analysis in the new version of the ms (lines 530-542).

Unfortunately, there is no experimental evidence that the ATPase from *Cyanothece* can really use sodium gradient.

With all due respect, we strongly disagree with your comment. After confirming that ATP depletion could be the reason for the failure of N₂ fixation in the absence of Na⁺, and recognising the striking resemblance between N₂-fixing *Cyanothece* sp. and anaerobic bacteria with Na⁺ energetics, as both require non-oxic microenvironments, we next directly used traditional experimental approaches widely applied in other bacterial Na⁺ energetics studies (McMillan et al., 2011; Doello et al., 2021), to directly investigate the existence of Na⁺ energetics during N₂ fixation.

Specifically, the results, including quantification of cellular ATP levels after NaCl addition and the effect of the Na⁺ ionophore monensin on the diazotrophic population, directly demonstrate that *Cyanothece* sp. cells must rely on the Na⁺ gradient to generate ATP for N₂ fixation, supporting the Na⁺ energetics proposal. These results are shown in Fig. 4d-4f with a full section of interpretation (lines 283-324 of the original submitted article file). In addition, in the new version of the manuscript, we have provided further evidence in support of the Na⁺ energetics proposal. Specifically, 1) we

quantified the cellular ATP content and the ATP/ADP ratio of non-N₂-fixing cells grown in BG11₀ (for seven days) following the addition of NaCl (18 g/L) in the dark. The addition of NaCl resulted in a significantly higher level of ATP and a higher ATP/ADP ratio compared to the control without NaCl addition (Fig. 5a, b), indicating that Na⁺ plays a critical role in driving ATP synthesis; 2) to further confirm the involvement of Na⁺ in ATP synthesis during N₂ fixation, dark phase N₂-fixing cells grown with NaCl were treated with monensin (a Na⁺-specific ionophore) or ethyl-isopropyl amiloride (EIPA, an inhibitor of Na⁺ channels and Na⁺/H⁺ antiporters) (Doello et al., 2021). The addition of monensin (14 μM) resulted in significantly lower ATP levels and ATP/ADP ratios compared to the untreated control (Fig. 5c, d). Similarly, treatment with EIPA (100 μM) markedly suppressed ATP synthesis, resulting in lower ATP/ADP ratios (Fig. 5e, f). These results substantiate the key role of Na⁺ energetics in N₂ fixation. Detailed changes can be found in lines 364-418.

Most ATPases are localised at the thylakoid membrane, where a proton motive force is maintained. They propose that an enigmatic Rnf complex generates a sodium motive force in the night by a sort of anaerobic respiration,

First, based on the transcriptomic data, we proposed two potential Na⁺ gradient generators (Rnf and Na⁺/H⁺ antiporter), the Na⁺/H⁺ antiporter should not be ignored. We sincerely appreciate the critical comments on Rnf. After a more careful review of the transcriptomic data, we find that our current transcriptomic evidence cannot conclusively support that the model organism harbours a Na⁺-pumping Rnf complex, as some subunits of Rnf complexes show similarities to subunits of the photosystem I complex (as you suggested). Therefore, we have removed Rnf from the manuscript. Furthermore, the transcriptomic patterns of the Na⁺/H⁺ antiporter genes show that all four Na⁺/H⁺ antiporter subunit encoding genes were significantly upregulated by non-N₂-fixing cells grown in BG11₀, compared to that of N₂-fixing cells grown in BG11₀ (NaCl) (Supplementary Fig. 7). This indicates that these cells struggled to meet their energy demands by attempting to build the transmembrane Na⁺ gradient and thus by Na⁺ energetics. The new results from the EIPA assay (an inhibitor of sodium channels

and Na⁺/H⁺ antiporters) further highlight the importance of the Na⁺/H⁺ antiporters in generating the Na⁺ gradient and subsequent ATP production (Fig. 5e, f). Detailed changes can be found in lines 446-460.

...without clarifying what this anaerobic respiration may be. In the discussion, the authors confuse anaerobic respiration with fermentation.

We apologise for confusing anaerobic respiration and anaerobic fermentation in the discussion section, the corresponding parts have been corrected (lines 611 - 620). However, as shown in Supplementary Fig. 7 and lines 425 - 445 in the original submitted file, we have proposed that anaerobic fermentation energy metabolism coupled with Na⁺ energetics powers N₂ fixation based on the results of expression of three key fermentative genes and activities of corresponding enzymes. Furthermore, by combining our results with published data (including the real-time cellular oxygen dynamics data during N₂ fixation that you strongly suggested), we are convinced that a mixed-acid type fermentation is coupled to Na⁺ energetics. Specifically, in the case of *Cyanothece* sp., it has been shown that cells can fix N₂ anaerobically (Liberton et al., 2019). Here, cells can generate energy in the absence of O₂, suggesting the existence of anaerobic energy metabolism. In addition, the N₂ fixation activity of anaerobically grown cells was no worse than that of aerobically grown cells, suggesting that cells may rely on anaerobic energy metabolism to fix N₂ even under aerobic conditions. Otherwise, the N₂ fixation activity of cells under aerobic conditions should be higher than that of cells grown without O₂, as it produces more ATP than anaerobic energy metabolism (Atteia et al., 2013). Moreover, cellular O₂ dynamics during N₂ fixation under aerobic conditions show that although there is a sharp decrease in the O₂ level of the culture medium, suggesting active aerobic respiration, this only occurs at the beginning of the dark phase. In contrast, the O₂ level remains almost unchanged, albeit with some perturbations (implying that almost no O₂ is consumed by the cells), and active N₂ fixation was simultaneously detected during the remaining hours (indicating a functioning energy metabolism) (Bandyopadhyay et al., 2013; Bandyopadhyay et al., 2024). This suggests that anaerobic energy metabolism, rather

than aerobic respiration, is operating. Indeed, it has already been shown that diazotrophic cyanobacteria rely on anaerobic fermentation to drive N₂ fixation under dark and anaerobic conditions (Heyer et al., 1989; Steunou et al., 2006). In particular, *Cyanothece* sp. contains all the fermentation-related genes necessary for the production of ethanol, lactate and acetate in its genome (Welsh et al., 2008), indicating the capability for anaerobic fermentation. Therefore, we investigated whether fermentation powers the N₂ fixation of *Cyanothece* sp.. We first examined the transcriptomic level of key fermentation-related genes. Although our transcriptomic analysis was designed to distinguish non-N₂-fixing growth-arrested cells from normally growing N₂-fixing cells at the transcript level, we identified interesting transcript patterns of fermentation-related genes suggesting that fermentation is involved in N₂ fixation. Specifically, we found that three key genes involved in microbial fermentation (Stal and Moezelaar, 1997), namely *ack* encoding acetate kinase, *adh* encoding alcohol dehydrogenase and *ldh* encoding lactate dehydrogenase, were upregulated in non-N₂-fixing cells compared to N₂-fixing cells (Supplementary Fig. 8a). This suggests that non-N₂-fixing cells may be challenged to meet their energy requirements in the absence of Na⁺ and therefore express higher levels of key fermentative energy metabolism genes. The presence of fermentative energy generation in N₂-fixing cells is further supported by the significantly higher activities of all three fermentative enzymes in N₂-fixing cells than in cells grown with N (Supplementary Fig. 8b). Moreover, based on the activities of three different fermentative enzymes, it is reasonable to conclude the existence of a mixed-acid type fermentation, a typical fermentation mode in bacteria (Stal and Moezelaar, 1997; Müller, 2008). Overall, it is highly likely that in the N₂-fixing *Cyanothece* sp. under dark and anaerobic conditions, aerobic respiration only plays a role during the initial period of the dark phase, probably to power the synthesis of nitrogenase and to create an anoxic interior, whereas mixed-acid fermentation coupled with Na⁺ energetics functions to power N₂ fixation during the resting hours. We have implemented the changes into the manuscript (lines 461-514).

Anaerobic respiration requires an alternative electron acceptor instead of oxygen. What should this be? It is generally accepted that aerobic respiration is a mechanism to deplete intracellular oxygen levels and create an anoxic environment in the cells. This has been clearly established in heterocysts.

As shown in other cyanobacterial fermentation studies, the electron acceptor of fermentation can be pyruvate/acetyl-CoA, organic intermediates of glycolysis (Müller, 2008). We have added this information to the manuscript (lines 630-633).

With regard to N₂-fixing heterocystous cyanobacteria, it is indeed widely accepted that aerobic respiration is a key mechanism to deplete intracellular oxygen levels and to create an anoxic environment. It is also known that heterocysts have special cell wall envelope structures (preventing O₂ diffusion) and lack the PSII photosystem (no O₂ production) to maintain the anoxic interior. However, the role of aerobic respiration in creating an anoxic intracellular environment does not necessarily mean that aerobic respiration must be employed or be the main energy source for subsequent N₂ fixation. For instance, if these specified cells use a combined strategy to create and maintain an anoxic interior, where does the O₂ come from if they have to rely on aerobic respiration to generate ATP for N₂ fixation? In other words, active N₂ fixation would require continuous or even high levels of O₂ uptake, which seems to be at odds with their cellular design. Although it is sometimes suggested that O₂ uptake and aerobic respiration must be tightly controlled, we believe that much more experimental effort is needed before the true role of aerobic respiration in heterocysts is fully understood. Indeed, many heterocystous cyanobacteria can only fix N₂ in the light, suggesting an important role for PSI photosynthesis in driving this process (Stal, 2015), otherwise heterocysts would continue to fix N₂ in the dark if they could obtain sufficient energy from aerobic respiration.

Whether the cells perform aerobic respiration or anaerobic respiration or anaerobic fermentation is solely dependent on the external availability of oxygen, or the presence or absence of other terminal electron acceptors, but this is no “free choice” of the bacteria, as the author insinuates.

As N₂ fixation takes place in a dark and anoxic environment (the latter created by the N₂-fixing cells themselves), we believe that it is likely that cells will switch from aerobic metabolism to anaerobic metabolism, even though in an aerobic environment, especially in the context of cells that they must rely on Na⁺ energetics. We have never intended to give the impression that the bacteria actively “choose” between the two processes. We have carefully reviewed the language in the manuscript and rephrased parts that might give this impression.

Further to this odd part of the manuscript comes the entire chaotic discussion about the seasonal fluctuations in salinity and temperature in coastal waters as an explanation for low N₂ fixing activities. When you look carefully at the data (provided in supplemental Fig 8), the contrary of what the authors claim seems to be the case: the lowest salinity is in April/May, whereas at the end of summer, in August, September, where the temperatures are highest, salinity is also high. So there is no antagonism between temperature and salinity. The entire explanation that growth supporting high temperatures coincide with low salinity, therefore impeding N₂ fixation is obsolete.

Thank you for your critical comments. The original intention of this analysis is to show a general pattern between coastal salinities and temperatures, which may help to interpret the relatively low abundance of *Cyanothece* sp. in nature. However, since we do not know other important abiotic and biotic information, such as the concentrations of combined N and P, the presence or absence of predators, and more importantly, the *in situ* abundance of *Cyanothece* sp., it will be very difficult to mimic the real situation in nature. After careful consideration we decided to remove this part from the manuscript, as you suggested elsewhere in the comments.

To me, the entire study seems to be preliminary and strongly biased by the belief of the authors in their sodium hypothesis. There are alternative explanations for the sodium dependence that have not been tested: as N₂ fixation occurs in the night, the cells need high amounts of glycogen to fulfill the energetic need of nocturnal N₂ fixation. If glycogen synthesis is dependent on increased sodium concentrations (for

example because of sodium-dependent bicarbonate transporters), then the sodium requirement could be indirect.

We respectfully disagree. Although one could strengthen parts of the study by performing more experiments (which we have done for this revised version of the manuscript – according to your suggestions), we think that our study is not preliminary but of general interest, as we have shown exciting results from many, carefully-designed and controlled experiments and provided corresponding interpretation based on the current literature. Specifically, 1) contrary to the traditional view that N₂ fixation of coastal cyanobacteria is sensitive to high salinity, we show the interesting phenomenon that *Cyanothece* sp., a coastal isolate, relies on NaCl to fix N₂; 2) with experimental evidence, we show step by step that cells must use Na⁺ energetics to produce ATP for N₂ fixation. These results not only demonstrate for the first time the critical role of Na⁺ energetics in N₂ fixation, but also provide rare experimental evidence for the existence of Na⁺ energetics in cyanobacteria; 3) by providing evidence that anaerobic mixed-acid fermentation is coupled to Na⁺ energetics to power N₂ fixation, we challenge the critical role of aerobic respiration in powering N₂ fixation.

Measuring the glycogen levels under the different growth conditions during day-night cycles is absolutely required.

Thank you for your constructive suggestion. Our results showing that *Cyanothece* sp. cells can grow and divide normally in BG11 (Fig. 1a) indicating that bicarbonate uptake, photosynthesis and subsequent glycogen storage are independent of Na⁺. Nevertheless, we agree with you that evaluating the effect of Na⁺ on cellular glycogen levels would provide more direct evidence as to whether Na⁺ is involved in bicarbonate uptake and subsequent glycogen metabolism. We have therefore carried out these experiments and the results support the idea that Na⁺ is not involved in bicarbonate uptake and glycogen biosynthesis during N₂ fixation (Fig. 4a). Detailed changes are provided in the updated manuscript (lines 289-306).

Another pivotal requirement is to determine the oxygen concentration or oxygen

requirement for nocturnal N₂ fixation. If the hypothesis of an anaerobic Rnf-complex based energy generation system is correct, then N₂ fixation would be optimal under anaerobic conditions. The consumption of oxygen during nocturnal N₂ fixation needs also to be determined.

Thank you very much for this suggestion, we believe that these data will provide pivotal evidence for the Na⁺ energetics-coupled anaerobic energy metabolism proposal. As a model organism, the cellular O₂ dynamics of the strain during N₂ fixation has been extensively studied (Bandyopadhyay et al., 2013; Bandyopadhyay et al., 2024). We have therefore decided to cite published data and conclusions from these studies. Detailed explanations can be found in the new version of the ms (lines 477-485).

Instead, the authors can skip the entire speculative paragraph on sodium dependent ATPase (without showing corresponding data) or the confusing discussion on the ecological consequences.

As clarified above in detail, we are confident that the AtpH sequence alignment analysis contributes to the topic of Na⁺ energetics in the cyanobacterial phyla and decided to respectfully reject this suggestion.

Regarding the comment "Skip the ecological consequence part". As explained above, this part has been removed as suggested.

A few other comments:

The description of experimental parameters is confusing. N and P are no nutrients, but are elements. The nutrients are N-compounds such as nitrate, ammonium.... or phosphorous. The concentrations of the nutrients should be indicated.

We have reviewed the entire manuscript and rephrased the relevant contexts (lines 176-177). In addition, the concentrations of the nutrients were indicated throughout the revised manuscript.

It is completely unclear how ATP should be taken up by the cells.

You are right that we do not know yet how extracellular ATP can be used by cells. To the best of our knowledge, there are very few studies investigating this topic. As you can see in the manuscript, we have used a combination of experiments to understand why cells cannot fix N_2 in the absence of Na^+ and showed that cells depend on Na^+ gradients for energy generation for N_2 fixation (Fig. 5). Overall, although it is interesting to understand how supplemented ATP is used by N_2 -fixing cells, we think it is not a must in this case and will not undermine our main conclusions as they are supported by other experiments in this study.

Under N-poor conditions, it will be used as a nitrogen source.

As ATP contains a nitrogen element, it is possible that ATP can act as a nitrogen source, which is why we have provided further evidence from two other experiments, indicating that the ATP additive here acts as extra energy. We are sorry that you somehow missed this information. Specifically, to distinguish whether ATP functions as extra energy or as a nitrogen source, we performed two more experiments, as shown in Fig. 4c and Fig. 4d, we found that cells grown in BG11₀+ATP synthesised a significantly higher amount of nitrogenase compared to cells grown in BG11₀+nitrate (almost no nitrogenase was detected), this result directly shows that ATP is not functioning as a nitrogen source here, otherwise cells should not synthesise nitrogenase. In addition, the results of the ATP, ADP and AMP addition experiment show that populations with ATP addition had a significantly higher cell density than those with ADP or AMP addition. Since equivalent doses of three additives have the same amount of nitrogen source but contain different amounts of energy, the differences in population density also indicate that ATP addition acts as energy but not as a nitrogen source. Therefore, from these two experiments, we believe that ATP supplementation does not function as N source.

As it was supplied at concentrations of 0.1 mM (according the methods), this cannot be compared to the 17 mM nitrate in standard BG11.

Thanks for your comment. In fact, we conducted this experiment several times, and for the first time, we used BG11₀ with NaCl (310 mM), BG11₀ with glutamine (2 mM), BG11₀ with ATP (0.1 mM) and standard BG11 medium, as described by the original method. However, we later found that all additives should be used with the same concentration otherwise it would be difficult to interpret the results, because the synthesis of nitrogenase from the ATP treatment could result from the depletion of ATP (similar to the depletion of NaNO₃, here ATP may function as N source). As a consequence, we re-conducted the experiments, in which we used BG11₀ with NaCl (310 mM), BG11₀ with glutamine (0.1 mM), BG11₀ with ATP (0.1 mM) and BG11₀ with NaNO₃ (0.1 mM). Although we used the same concentration of additives for the experiments, we are sorry that we did not correct our methods before submission and for the subsequent confusion. New method is described in lines 857-862.

There are several mistakes and confusions in the text, just to mention some of them:

Further specific points

l. 48: many bloom formers in fresh water are non-diazotrophic (e.g. Microcystis).

Microcystis blooms in fresh water. In the original manuscript, we did not claim that only N₂-fixers can bloom in freshwater ecosystems. Since the focused ecological paradox contrasts the performance of N₂-fixing cyanobacteria between freshwater lakes and marine coastal waters, non-N₂-fixing cyanobacterial bloomers were excluded from the original context. To avoid confusion, we rephrased these sentences which you can find in line 48.

l. 60: high salinity doesn't generally suppress N₂ fixation in filamentous cyanobacteria. For example, Trichodesmium is actively fixing N₂ in seawater.

The ecological paradox described at the beginning of this manuscript focuses on the question of why coastal marine ecosystems are usually N-limited while dwelling N₂-fixing cyanobacteria cannot prosper, even though they have a clear ecological

advantage (Howarth and Marino, 2006; Conley et al., 2009). The focus of this study is therefore on coastal ecosystems, which is known for their dynamically changing salinities. It is true and interesting that open ocean species such as *Trichodesmium* sp. and the UCYNA and UCYNB groups of cyanobacteria can fix N₂ in the presence of NaCl, or even have to rely on it, but they do not fit the research focus of this study, i.e., the coastal ecosystem, which is where *Cyanothece* sp. lives.

I. 71-75: here you turn the paradox upside down without solving it

The first part of this study is to understand the ecological paradox of low abundance of N₂-fixing cyanobacteria in normally N-limited coastal waters. As previous studies mainly focused on multicellular N₂-fixers and reported that N₂ fixation of multicellular N₂-fixers is sensitive to high salinities (> 10 g/L), we aimed to understand the paradox from the perspective of unicellular N₂-fixers. Although, in contrast to these studies, we reported that the model organism *Cyanothece* sp. must rely on NaCl to fix N₂, we would not claim to turn the paradox upside down. Rather, we prefer to say that we have provided another angle to reconsider the role of NaCl in coastal N₂ fixation. We think that our results are interesting, especially in the context of unicellular cyanobacterial N₂-fixers, which are increasingly recognised for their significant contribution to the oceanic N cycle. Furthermore, as explained above, we believe that our study reports an interesting discovery and provides detailed insights into the underlying mechanisms.

The I- 143- 150: Here the authors confuse stringent response with nitrogen-starvation induced chlorosis. In all microbiology textbooks, stringent response is presented as the specific response mediated by the alarmone ppGpp. In heterotrophic bacteria, amino acid starvation is the most important trigger for elevated ppGpp and thus of the stringent response. However, in cyanobacteria, this is completely different: Here ppGpp coincides with nocturnal dormancy: the lack of energy in the absence of photons causes elevated ppGpp, as clearly shown for *Synechococcus* (Hood, Higgins et al., PNAS 2016 and similar reports). By contrast, nitrogen chlorosis is induced by a complex network of stress-responsive transcription factors, such as NblS or NtcA,

which ultimately initiate chlorosis. This has been thoroughly elaborated in *Synechocystis*, see several recent reviews on this topic.

First, during our analysis of the transcriptomic data (N_2 -fixing cells with NaCl vs. non- N_2 -fixing cells without NaCl), we found that non- N_2 -fixing cells: 1) were N-starved and growth-arrested (typical cause and consequence of the stringent response); 2) showed a downregulated expression pattern of ribosome biosynthesis genes but an upregulated expression pattern of nutrient transporter genes (typical transcriptomic pattern of the stringent response); 3) upregulated the expression of ppGpp, a signalling molecule for the stringent response, encoding the gene *relA* (gene location: NC_010546.1 (21413 - 22405), NCBI). All three pieces of evidence perfectly meet the criteria for defining the bacterial stringent response (Irving et al., 2021), and we have not seen any reason why it is inappropriate to draw the stringent response conclusion.

Regarding the article you suggested, the PNAS article is titled “The stringent response regulates adaptation to darkness in the cyanobacterium *Synechococcus elongatus*”, and the main conclusion is “...we show that the stringent response - a stress response pathway whose enzymes are conserved in nearly all bacteria, as well as plant plastids - is involved in dark adaptation in *Synechococcus*.” So we are confused as to what you mean by “However, in cyanobacteria, this is completely different: Here ppGpp coincides with nocturnal dormancy: the lack of energy in the absence of photons causes elevated ppGpp, as clearly shown for *Synechococcus* (Hood, Higgins et al., PNAS 2016 and similar reports).”. Do you mean that ppGpp in cyanobacteria are not involved in the stringent response or that only darkness can induce the cyanobacterial stringent response? Because for us, this PNAS paper is a study of the cyanobacterial stringent response and the ppGpp identified are involved in the stringent response, although the inducer is not the typical amino acid starvation but darkness. Furthermore, in addition to darkness, as stated in many other publications, e.g., Hauryliuk et al., 2015, the bacterial stringent response can be induced by a number of environmental factors, such as nutrient limitation, heat shock and the presence of antibiotics. Overall, we believe the conclusion of the stringent response in the manuscript is reasonable.

Regarding the question of the stringent response or nitrogen chlorosis. As far as we know, the stringent response is a ubiquitous stress signalling pathway in response to nutrient starvation in bacteria, whereas chlorosis usually describes the processes involved in the acclimation of non-diazotrophic cyanobacteria to nitrogen shortage (Forchhammer & Schwarz, 2019). In our case, although *Cyanothece* sp. is able to fix N₂, the cells lost this ability in the absence of Na⁺ and were thus N-starved. Therefore, we believe that the phenomenon we observed can be classified as both stringent response and chlorosis, two theories that are not in conflict with each other. We have included chlorosis in the updated manuscript, and detailed changes can be found in lines 162-170, 301-303, and 602-604.

L. 264. Here you need to mention the concentrations of Gln or nitrate.

Information added as suggested, now line 309.

L. 360: Sentence is unclear. Do you mean “except its role” instead of “but see its role”

Modified as suggested (line 524).

L. 371: What is “Low+Medium/High”? Do you mean, “low, medium or high”?

Based on the AtpH sequence data, we find that some species have two AtpH homologs, one of which can be defined as low H⁺ selectivity. The other can be defined as either medium or high H⁺ selectivity. More specifically, species with low H⁺ selectivity always have two AtpH homologs - one with low and the other with medium or high H⁺ selectivity. So, Low+Medium/High actually means Low+Medium or Low+High. To make this clearer, we have changed the contexts and labels of the figure. Detailed changes can be found in lines 534-535 and 574-576.

Generally:

In the first part of the paper, it should be clearly stated how the cells were grown in diurnal cycles, and that N₂ fixation only occurs during the dark period.

Modified as suggested. Detailed changes can be found in lines 280-281.

The role of cyanophycin as transient nitrogen storage compound has been completely neglected.

Thanks you for suggesting cyanophycin. In *Cyanothece* sp., it is known that cells store combined nitrogen in the form of cyanophycin and carbon hydrate in the form of glycogen. Since the focus of this study is to understand why N₂ fixation fails in the absence of Na⁺, we do not believe that cyanophycin plays an important role in the subsequent process of successful N₂ fixation. In fact, in our experimental designs, to avoid the confounding effect of cyanophycin, which can support short-term population growth, we sampled and tested 7-day-old populations, by which time we believe the stored combined nitrogen has been depleted and the increase in population growth must rely on N₂ fixation. Nevertheless, we believe the inclusion of cyanophycin will make the manuscript more complete, so we have decided to include cyanophycin in the latest manuscript. Detailed changes can be found in lines 87-91.

Concerning the transcripts that you have investigated: In the databases, no rnf gene is annotated for *Cyanothece*. Provide the accession numbers of the genes that you mention in this study, otherwise, it is not clear what your really looked up. Some subunits of Rnf complexes show similarities with subunits of Photosystem I complex. What is the evidence that this gene you termed “rnf” is really part of an Rnf complex?

Regarding the suggestions for genes with accession numbers, we have modified this as suggested throughout the manuscript.

Regarding the comment on “rnf gene”. Removed and clarified as above.

Reviewer #3 (Remarks on code availability):

I cannot access this site.

[DOI or URL: <https://github.com/SiTANG1990/N2-fixation-coastal-unicellular-cyanobacteria.git>.]

We apologise, this has now been updated.

References:

1. Irving, S. E., Choudhury, N. R. & Corrigan, R. M. The stringent response and physiological roles of (pp)pGpp in bacteria. *Nat Rev Microbiol* **19**, 256–271 (2021).
2. Hauryliuk, V., Atkinson, G. C., Murakami, K. S., Tenson, T. & Gerdes, K. Recent functional insights into the role of (p)ppGpp in bacterial physiology. *Nat Rev Microbiol* **13**, 298–309 (2015).
3. Forchhammer, K. & Schwarz, R. Nitrogen chlorosis in unicellular cyanobacteria – a developmental program for surviving nitrogen deprivation. *Environmental Microbiology* **21**, 1173–1184 (2019).
4. Von Ballmoos, C., Wiedenmann, A. & Dimroth, P. Essentials for ATP Synthesis by F₁ F₀ ATP Synthases. *Annu. Rev. Biochem.* **78**, 649–672 (2009).
5. Doello, S., Burkhardt, M. & Forchhammer, K. The essential role of sodium bioenergetics and ATP homeostasis in the developmental transitions of a cyanobacterium. *Current Biology* **31**, 1606-1615.e2 (2021).
6. McMillan, D. G. G. et al. A1Ao-ATP Synthase of *Methanobrevibacter ruminantium* Couples Sodium Ions for ATP Synthesis under Physiological Conditions. *Journal of Biological Chemistry* **286**, 39882–39892 (2011).
7. Liberton, M., Bandyopadhyay, A. & Pakrasi, H. B. Enhanced Nitrogen Fixation in a glgX -Deficient Strain of *Cyanothece* sp. Strain ATCC 51142, a Unicellular Nitrogen-Fixing Cyanobacterium. *Appl Environ Microbiol* **85**, e02887-18 (2019).
8. Atteia, A., Van Lis, R., Tielens, A. G. M. & Martin, W. F. Anaerobic energy metabolism in unicellular photosynthetic eukaryotes. *Biochimica et Biophysica Acta (BBA) - Bioenergetics* **1827**, 210–223 (2013).
9. Bandyopadhyay, A., Sengupta, A., Elvitigala, T. & Pakrasi, H. B. Endogenous clock-mediated regulation of intracellular oxygen dynamics is essential for diazotrophic growth of unicellular cyanobacteria. *Nat Commun* **15**, 3712 (2024).
10. Bandyopadhyay, A., Elvitigala, T., Liberton, M. & Pakrasi, H. B. Variations in the Rhythms of Respiration and Nitrogen Fixation in Members of the Unicellular Diazotrophic Cyanobacterial Genus *Cyanothece*. *Plant Physiology* **161**, 1334–1346 (2013).
11. Welsh, E. A. et al. The genome of *Cyanothece* 51142, a unicellular diazotrophic cyanobacterium important in the marine nitrogen cycle. *Proc. Natl. Acad. Sci. U.S.A.* **105**, 15094–15099 (2008).
12. Heyer, H., Stal, L. & Krumbein, W. E. Simultaneous heterolactic and acetate fermentation in the marine cyanobacterium *Oscillatoria limosa* incubated anaerobically in the dark. *Arch. Microbiol.* **151**, 558–564 (1989).
13. Steunou, A.-S. et al. In situ analysis of nitrogen fixation and metabolic switching in unicellular thermophilic cyanobacteria inhabiting hot spring microbial mats. *Proc. Natl. Acad. Sci. U.S.A.* **103**, 2398–2403 (2006).
14. Stal, L. J. & Moezelaar, R. Fermentation in cyanobacteria. *FEMS Microbiology Reviews* (1997).
15. Müller, V. Bacterial Fermentation. in eLS (John Wiley & Sons, Ltd, 2008).

doi:10.1002/9780470015902.a0001415.pub2.

16. Stal, L. J. Nitrogen Fixation in Cyanobacteria. in eLS (ed. John Wiley & Sons, Ltd) 1–9 (Wiley, 2015).
17. Howarth, R. W. & Marino, R. Nitrogen as the limiting nutrient for eutrophication in coastal marine ecosystems: Evolving views over three decades. *Limnology and Oceanography* 51, 364–376 (2006).
18. Conley, D. J. et al. Controlling Eutrophication: Nitrogen and Phosphorus. *Science* 323, 1014–1015 (2009).

Reviewer #4:

This manuscript reports the important relationship between N₂ fixation and Na⁺ energetics in unicellular coastal cyanobacteria. This is an interesting study concerning cyanobacterial physiology and is of interest from the view of the global marine N-cycle. Although comprehensive studies have been conducted, this reviewer would advise the authors to consider the following points.

Thank you very much for your enthusiasm for our work and evaluation of the manuscript. We have modified the manuscript according to your suggestions.

General comments

The following two points are important and should be considered by the authors first.

1. Growth phenotypes shown in Figure 1 clearly show the strong effect of NaCl on both the N₂ fixation and the growth of *Cyanothece*. Then the authors focused on the nitrogenase activity and the amounts of ATP in the cell which may affect the N₂ fixation function. To confirm if a limitation of ATP led to N₂ fixation, they supplemented ATP into the culture medium and observed the stimulation of the growth. This is a very interesting result. However, since nothing is known about ATP transporters or carriers on the cell membrane, the interpretation of this result is a major issue. If supplementation of ATP in the medium increases the ATP content in the cell, the authors must first ascertain whether this ATP effect is simply the uptake of ATP from the medium or some signaling effect of ATP on the cell. In addition, the energy level in the cell is not determined solely by the amount of ATP in the cell even if the supplemented ATP is actively taken up by the cell. The balance between ATP and ADP is critical, and the authors must at least directly determine the amounts of adenylates in the cell.

You are right that we do not know yet how extracellular ATP can be used by cells. To the best of our knowledge, there are very few studies investigating this topic. As you can see in the manuscript, we have used a combination of experiments to understand

why cells cannot fix N_2 in the absence of Na^+ and showed that cells depend on Na^+ gradients for energy generation for N_2 fixation (Fig. 5). Overall, although it is interesting to understand how supplemented ATP is used by N_2 -fixing cells, we think that understanding the underlying mechanism is not a required in our study as it will not affect our main conclusions as they are supported by other experiments. Nevertheless, we still chose to show the data as we are convinced that despite the fact that we do not quite understand how ATP is used by the cells, the results are of significance and together with the ADP and AMP experiments clearly demonstrate that somehow the cells are capable of harnessing the energy.

Thank you for your suggestion regarding ADP quantification. We followed your advice and quantified the cellular ADP level and calculated the ATP/ADP ratio. We first tested the effect of amiloride (EIPA) on cellular ATP levels and the ATP/ADP ratio of N_2 -fixing cells in the presence of Na^+ (Fig. 4e, f), the results show that treatment with EIPA (100 μ M) markedly suppressed ATP synthesis, resulting in a lower ATP/ADP ratio (Fig. 5e, f), substantiating the pivotal role of Na^+ energetics in generating ATP for N_2 fixation. We then tested the effect of EIPA on cell growth in different media, and the results show that no growth was observed in any of the treatments exposed to EIPA (100 μ M) (Fig. 6c), suggesting that EIPA inhibited both the Na^+/H^+ antiporter and other essential cellular functions through EIPA. The lower cellular ATP levels and ATP/ADP ratios observed following EIPA treatment (Fig. 5e, f) may be due not only to the inhibition of Na^+ -dependent ATP synthesis but also to the suppression of other key metabolic processes. Detailed changes can be found in lines 364-418.

2. The second half of the manuscript focuses on Na^+ energetics and Na^+ -driven ATP synthases. This is another very important perspective.

Unfortunately, the authors only focus on the amino acid sequence of the c-ring, which has already been studied in databases (Refs. 19, 32). On the other hand, it has already been reported and biochemically proven that several ATP synthases from bacteria living in specific environments are Na^+ gradient-driven (Dimroth et al.). However, this

is still a very rare case, and so far, nothing is known about cyanobacterial ATP synthases. Because the information provided by this manuscript is so important to understanding the physiology of cyanobacteria living in a Na⁺ environment, a solid biochemical basis would be needed to discuss cell growth by Na⁺-driven ATP synthase in this manuscript.

Thank you for your constructive comments. We are sorry for the confusion in the manuscript, which may have led to your misinterpretation of our results (if we understand your comments correctly). In fact, prior to the AtpH sequence analysis, in order to demonstrate Na⁺ energetics under N₂ fixing conditions, we provided several biochemical results (Fig. 4d – f of the original submitted file) to show that cells must rely on Na⁺ gradient to generate ATP for N₂ fixation, i.e., Na⁺ energetics. Specifically, we used traditional experimental approaches widely applied in other bacterial Na⁺ energetics studies (McMillan et al., 2011; Doello et al., 2021), to test for the presence of Na⁺ energetics during N₂ fixation. Specifically, our results, including the quantification of cellular ATP levels after NaCl addition and the effect of the Na⁺ ionophore monensin on the diazotrophic population, directly demonstrate that *Cyanothece* sp. cells must rely on an Na⁺ gradient to generate ATP to power N₂ fixation, thus supporting the Na⁺ energetics proposal. For the updated manuscript, we have performed further experiments to explore the relationship between Na⁺ and cellular ATP levels and ATP/ADP ratios as described above, and the results support the Na⁺ energetics proposal. Thus, we believe that we have provided biochemical evidence for the importance of Na⁺ energetics in N₂ fixation. To provide further experimental evidence for the Na⁺ energetics proposal, we conducted additional experiments, including the quantification of cellular ATP levels and ATP/ADP ratios (as you suggested) of: 1) non-N₂-fixing, growth-arrested cells (grown in BG11₀) after the addition of Na⁺; 2) N₂-fixing normally growing cells after the addition of monensin (Na⁺ ionophore) or EIPA (an Na⁺/H⁺ antiporter inhibitor). As shown in Fig. 5 & Fig. 6, non-N₂-fixing growth-arrested cells significantly increased cellular ATP levels and the ATP/ADP ratio upon the addition of NaCl, whereas N₂-fixing normally growing cells

significantly decreased cellular ATP levels and the ATP/ADP ratio upon the addition of monensin or EIPA, indicating the crucial role of Na⁺ in generating ATP, i.e., Na⁺ energetics. In addition, we used the AtpH sequence alignment analysis to explore the possibility that other cyanobacteria may also be capable of Na⁺ energetics (but not as key evidence for the existence of Na⁺ energetics for our model organism). We are asking how prevalent this phenomenon is across the cyanobacterial phyla. However, as you have commented, it will be very important and interesting to delve into the details of Na⁺ energetics in other cyanobacterial species. Nevertheless, we believe that our current evidence is sufficient to support our main point: Na⁺ energetics-coupled N₂ fixation, and we respectfully decide not to include your suggestion in the current study, but will consider it in our future projects.

Minor comments

1. Fig. 1a and b Since it is very difficult to set the same growth conditions, a logarithmic graph is usually used to show the growth curve.

We have modified this as suggested (see Fig. 1a and b).

2. Fig. 1c-e To find the most useful Na⁺ conditions, these graphs should be shown in scatterplots.

The purpose of Fig.1c-e is not only to show the most useful Na⁺ concentrations, but also to show that cells cannot fix N₂ even in the presence of low concentrations of Na⁺.

We therefore respectfully differ from your advice.

3. line 242. Information about the measurement of nitrogenase content in the cell is very limited. Though the authors described the method in the 'materials and methods' section (see lines 665-666), it is unknown how they can determine the amounts of the desired protein by commercial ELISA. At least, the purified recombinant protein must be necessary as a standard for this measurement.

Sorry for the confusion, but like many other methods, the ELISA method always

requires a standard curve to calculate the concentrations of the samples. We have provided more details in the methods section, detailed information can be found in lines 787-797. In fact, we were not sure about the accuracy of this method at the beginning, so we set up a nitrate treatment to test its accuracy. As shown in Fig. 3d, hardly any nitrogenase was detected in the nitrate treatments, suggesting that the accuracy of this method is acceptable.

4. line 278 and supplementary Fig. 4a: ADP and AMP must be also examined.

Thank you for your suggestion on ADP and AMP, we believe that the inclusion of AMP in this experiment will provide additional evidence that ATP functions as energy but not as a source of N. However, after more careful consideration, we think that Supplementary Fig. 4a of the original manuscript is not closely related to the topic of this section, as the result that the consumption or decay of ATP did not provide additional evidence that ATP functions as energy but not as an N source (if ATP were to function as an N source, similar patterns would also be expected), we decided to remove this part. Instead, we included AMP in Fig. 4d, as can be seen in lines 321-330. The results show that the ATP treatment has the highest population density, followed by the ADP treatment, while the addition of AMP leads to the lowest final density.

5. line 293: The authors should consider the possibility that Na^+ alters the H^+ gradients in the cell and indirectly affects the ATP synthesis reaction. Direct biochemical measurement is necessary to conclude.

Thanks for the interesting comment, it may be true that Na^+ alters the H^+ gradient and thus indirectly affects ATP synthesis. However, as shown in Fig. 6b, in the presence of protophore DTHB, which disrupts the transmembrane H^+ gradient, cells can normally fix N_2 and grow, suggesting that under N_2 fixation conditions, only Na^+ energetics is functional.

6. line 744: How can they determine the amounts of ATP in the cytosol by this method?

It should be described precisely.

As we followed your advice to quantify cellular ADP levels, we decided to re-quantify (and confirm) cellular ATP levels at the same time using another approach. The results show that, although there is variation in the absolute values, similar trends are observed between the different methods. Detailed information on both methods is given as suggested (lines 869-900).

8. Supplementary Fig. 1 Although the authors simply determined the chlorophyll content in the cells, spectral analyses might be more useful for studying changes in the various colored pigments in the cells.

The reason for quantifying cellular chlorophyll content is to show that non-N₂-fixing cells are N-starved in the absence of Na⁺, since chlorophyll contains the nitrogen element. Therefore, although obtaining more details of different coloured pigments may provide more details of the respective changes of each pigment, its contribution to our main conclusion is the same as what we have now, so we therefore decided not to go into these details.

7. lines 446-470: This reviewer wonders if a discussion of the relationship between observed salinity dynamics in the Texas Gulf Coast and cyanobacterial growth based on laboratory data is appropriate as a topic for a separate section in Results. The figures shown in Supplementary Figures 8a-c are based on published data, and only Figure 8d was generated from original data by the authors themselves. Therefore, interpretations relating to the two should be written in the discussion.

Thank you for your kind suggestion. In addition, since we do not know other important abiotic and biotic information, such as the concentrations of combined N and P, the presence or absence of predators, and more importantly, the *in situ* abundance of *Cyanothece* sp., it will be very difficult to mimic the real situation in nature. After careful consideration, to follow your advice and remove this part from the Results section and only discuss it briefly in the Discussion section. Detailed modifications are in lines 655-

References:

1. Doello, S., Burkhardt, M. & Forchhammer, K. The essential role of sodium bioenergetics and ATP homeostasis in the developmental transitions of a cyanobacterium. *Current Biology* **31**, 1606-1615.e2 (2021).
2. McMillan, D. G. G. et al. A1Ao-ATP Synthase of *Methanobrevibacter ruminantium* Couples Sodium Ions for ATP Synthesis under Physiological Conditions. *Journal of Biological Chemistry* **286**, 39882–39892 (2011).

Response to reviewers' comments

Reviewer #1 (Remarks to the Author):

The authors have addressed the reviewer's concerns.

The manuscript was revised properly and ready for the publication.

Thank you very much for your enthusiasm for this work and evaluation of the manuscript.

Reviewer #2 (Remarks to the Author):

The authors have addressed all of my questions.

Thank you very much for your enthusiasm for this work and evaluation of the manuscript.

Reviewer #3 (Remarks to the Author):

The paper by Tang et al has considerably improved as compared to the previous version. However, there are still important issues that need clarification:

1. Following sentence needs correction:

„They up-regulated the expression of ppGpp (Supplementary Fig. 4b, gene location: NC_010546.1 (21413 - 22405), NCBI), which is the signalling molecule for the stringent response encoding gene10.

The relA gene (NC_010546.1) does not encode ppGpp, but the ppGpp synthase/hydrolase enzyme. These enzymes are usually under a strict metabolic control (for example induced by non-aminoacylated tRNA bound to the ribosome). Therefore, induced expression of relA is not a direct proof of elevated ppGpp levels. To measure ppGpp directly, this metabolite needs to be quantified. The induced expression of relA can therefore only be taken as an indication of increased ppGpp levels.

Thank you for your informative comment, it helps a lot. We agree with your comment and have modified this part accordingly based on your suggestion. Detailed changes can be found in lines 154-157 of the revised manuscript.

2. I do not understand the logic of the following:

„As shown in Fig. 2c, elevated transcript levels of all four P limitation biomarkers were observed in these non-N₂-fixing cells, indicating that the cells were P-starved, likely as an indirect consequence of N starvation. This up-regulated pattern of these genes makes biological sense: starving cells struggle to meet nutrient demands by expressing higher levels of nutrient synthases or transporters¹¹“

The Pi assimilating genes are very specific for Pi limitation....It's very unlikely that N-limitation induces Pi assimilation genes (if I am wrong, provide a reference for such cross-talk) Moreover, growth arrested cells do not consume phosphorous since nucleic acid synthesis is also stopped. Therefore, the up-regulated pattern makes no immediate biological sense. There must be another reason for up-regulation of Pi assimilation genes.... perhaps the need to take up sodium is linked to Pi assimilation?

Thank you for your comment. The original underlying logic was that N-limitation led to

the failure of protein synthesis, including Pi transporter proteins. As a consequence, cells were P-starved and responded to such a situation via the upregulation of P-limitation genes. Now, we agree with you that “growth arrested cells do not consume phosphorous since nucleic acid synthesis is also stopped. Therefore, the up-regulated pattern makes no immediate biological sense”.

As to the possibility that sodium may be linked to Pi assimilation, we don't think this is the case in this study, as we show that Pi uptake is independent of the presence of Na⁺ (Fig. 1a & Fig. 3c), whereby cells supplied with combined N can normally grow and divide in the absence of Na⁺, which would otherwise lead to growth arrest if Pi uptake had to rely on the presence of Na⁺.

Overall, after careful consideration and discussion, we think that this part is not closely related to the main findings of this study and are concerned that this part may be confusing. We have therefore decided to remove the transcriptomic analyses of Pi limitation genes, including figures and corresponding context. Changes can be found in line 171.

3. Following point is unclear:

„This result, coupled with the fact that cells can grow normally in BG11 (Fig. 1a), suggests that Na⁺ is not involved in bicarbonate uptake and glycogen synthesis“

It is not specified, how cell were cultivated and supplied with inorganic carbon. Cells depend on active (sodium-dependent) bicarbonate uptake only at ambient CO₂ concentrations. With higher CO₂ supply, CO₂ can enter cells by diffusion.

Thank you for your kind comment. Since CO₂ uptake or CO₂ concentrating mechanisms are not the focus of this study, we cultured cells without forced aeration, the only CO₂ supply being diffusion from the atmosphere into the flasks. Corresponding information has been provided in lines 282, 605-606 of the revised manuscript.

4. How the cells make use of extracellular ATP is still a mystery. If they would take it up, they would also use the nitrogen, as the purines are a rich source of combined N. What would happen to all the ADP/AMP from ATP turn-over, once the high energy gamma-phosphate of ATP has been consumed? According to the authors hypothesis, it cannot be re-charged by ATPase reaction....

Thank you for your comment. To the best of our knowledge, there are few studies investigating the underlying mechanisms of extracellular ATP uptake by bacteria. Although results show that several bacteria can consume extracellular ATP (Mempin et al., 2013), which means that an ATP utilisation/uptake mechanism must exist, the uptake mechanism remains unclear. Possibilities include that ATP may be transported into cells via water-filled porins (at least into the periplasm) or via transporters/carriers or other uptake systems, either independently or in combination, and then utilised by cells. Future experimental evidence is required to address this issue. We have discussed this in the new version of the manuscript (lines 322-325).

Before conducting two additional experiments (the nitrogenase quantification and the ATP, ADP, AMP addition experiments, Fig. 4c, d), we also wondered whether ATP could potentially serve as N source. However, the results of these experiments indicate that added ATP acts as energy but not as N source, because cells synthesised nitrogenase in the presence of ATP (Fig. 4c) and equivalent doses of ATP, ADP and AMP led to different increases in cell number (Fig. 4d). If it was a source of N, one would have expected the same increase in cell number for ATP, ADP, AMP as the nitrogen content is the same in all molecules.

You are correct that the ADP/AMP from the ATP turnover are not recharged to form ATP under these growth conditions, as we have shown that, unlike NaCl addition, the batch addition of exogenous ATP cannot support continuous population growth due to the depletion of ATP by cells and the ATP decay in the medium over time (Supplementary Fig. 4a of the originally submitted Supplementary Information, see the attached figure below). However, in our latest revised manuscript, we deleted this figure and the corresponding context because we found that either function as an energy or N source, the depletion or decay of the ATP supplement would lead to a similar growth pattern (shown in the figure below).

Supplementary Fig. 4a. Growth curves of cells grown in BG11₀ or supplemented with either NaCl or ATP over a period of 12 days.

5. Concerning anaerobic N₂-fixation:

„In fact, *Cyanothece* sp. cells have been previously shown to fix N₂ anaerobically²⁷.

Here, cells can generate energy in the absence of O₂, suggesting the existence of an anaerobic energy metabolism. In addition, the N₂ fixation activity of anaerobically grown cells was no worse than that of aerobically grown cells....,“

I am afraid that the authors misunderstood Ref. 27, which they give as a reference for anaerobic diazotrophy... In this work, only the nitrogenase activity assay was done under anaerobic conditions (flushing cells with argon) but not cell growth: Anaerobic nitrogenase assay is a routine assay to check if the oxygen protection system of nitrogenase is ok. However, the cells have not been cultivated under anaerobic conditions. If the authors want to prove their hypothesis of an anaerobic fermentative metabolism that supports nitrogenase, then they need to show it experimentally.

Thank you for pointing out these details of ref. 27. You are quite right that their results can only show that cells are capable of fixing N₂ anaerobically and cannot conclusively support that *Cyanothece* sp. can fix N₂ for growth under anaerobic conditions. The quantification of population growth or a direct quantification of the reduction of gaseous N₂ under anaerobic conditions is needed to support such a claim. As a result, we have

removed this reference and related context. But we think the remaining evidence in that section still strongly suggests that anaerobic fermentation plays a crucial role in the energy generation for N₂ fixation.

By now, to our best knowledge, there is no experimental evidence in support of that *Cyanothece* sp. cells can actually grow fixing N₂ under strictly anaerobic conditions. Honestly, it is a little bit tricky to address this question as *Cyanothece* sp. will produce O₂ via photosynthesis even though they are initially grown in sealed O₂-free flasks, and the self-produced O₂ will undermine the anaerobic environment.

Furthermore, as indicated by the title of the section “Anaerobic fermentation possibly powers N₂ fixation”, we have suggested that it is highly likely that anaerobic fermentation powers N₂ fixation, which is not yet a definitive conclusion. Although it would be great to conclusively understand the role of anaerobic fermentation in N₂ fixation, we think that the information provided in the manuscript is sufficient to support such a proposal. In addition, it will not undermine our main conclusion, i.e., Na⁺ energetics-dependent N₂ fixation, as this is supported by other experiments in this study. Changes in this part can be found in lines 482-485 and 518.

Reference:

1. Mempin, R. et al. Release of extracellular ATP by bacteria during growth. *BMC Microbiol.* **13**, 301 (2013).

Reviewer #4 (Remarks to the Author):

In the revised manuscript, the authors take the reviewers' comments seriously and have responded to them. However, the following points still require further consideration.

1. Na⁺-dependent ATP synthesis

The data presented by the authors certainly indicate that N₂ fixation is affected by Na⁺-energetics. However, as pointed out by other reviewers, it is undetermined whether Na⁺-dependent ATP synthesis is actually taking place. For this reason, placing Figs. 7b and c at the end of the Results seems misleading; perhaps they should be indicated in discussion.

Thank you for your kind comment. By far, the results presented already indicate that Na⁺ energetics play an essential role in the activation of N₂ fixation. Although we agree with you that further experimental efforts are needed to provide more direct evidence to determine whether and to what extent Na⁺-dependent ATP synthesis actually takes place, we have decided to address this issue in future work, as explained in lines 413-415 of the revised manuscript. After careful consideration and based on the suggestions from the editor, we decided not to move Fig. 7b, c to the discussion section. Instead we have toned down the language throughout the manuscript (line 462 and 550) and moved Fig. 7b, c to the section "Potential Na⁺ gradient generator", as you can find in lines 453-472.

2. Experimental conditions

The ion concentrations used in this study seem to be deterministic everywhere (e.g., L211, L217, L237, L317, L318, L322). An explanation may be in order.

Thank you for your kind suggestion. You are quite right that the ion concentrations used in this study were chosen for specific reasons. Firstly, the concentration of NaCl (310 mM, L211) was chosen based on the difference in NaCl between the artificial seawater medium ASP2 and the freshwater medium BG11. Since ASP2 without combined N (310 mM NaCl) can support the population growth of *Cyanothece* sp., whereas BG11 without combined N cannot (Fig. 1a), the presence or absence of NaCl (310 mM) was chosen for future tests. To investigate whether other metal chlorides

could mimic the effect of NaCl in stimulating N₂ fixation and subsequent population growth, the same concentration (310 mM) was used for all treatments, including LiCl (L217).

Secondly, the concentration of NH₄Cl and Gln (2 mM, L237) was decided based on the result of a pre-experiment in which a gradient of both additives was tested for their ability to stimulate population growth. Based on this result, 2 mM was chosen as it is neither too low to be unable to support population growth nor too high to potentially inhibit population growth.

Thirdly, as the commercially available aqueous ATP solution has a concentration of 10 mM and the volume of BG11₀ medium used in the experiment is 10 mL, a final concentration of 0.1 mM ATP (L318) was chosen to alleviate the dilution of the culture medium by the addition of the ATP mother solution and to ensure a sufficiently high ATP concentration to support population growth, 100 µL of ATP mother solution was added to 9900 µL of each culture medium. As all treatments should have the same concentration of additives, 0.1 mM Gln and NaNO₃ (L317) were used in this experiment. Finally, for the same reason as above, 0.1 mM ATP was used and an equivalent dose of ADP and AMP (L322) was also used.

Detailed explanations for the use of specific ion concentrations can be found in lines 197, 700-703, 791-797, 802-804 and 810-811.

3. Notation of Cell Concentration

In this manuscript, cell volume is expressed in terms of cell number (L688). Since differences in experimental conditions (salt concentration) used at various parts may affect the size of cells, it may not be appropriate to use simply the number of cells for quantification.

Thank you for your comment. We acknowledge that experimental conditions may potentially affect cell size. However, in this manuscript we deliberately counted the number of cells and did not measure OD as a proxy for population size as OD could indeed be affected by cell size and thus not correctly represent cell/ population fitness. However, it should be noted that the growth trajectories and carrying capacities of populations cultured in either BG11 or ASP2 (different salinities, N replete) do not differ

(Fig. 1a), indicating that cell sizes are very similar. Furthermore, there does not appear to be a trade-off between cell number and cell size, as smaller populations also consist of cells with less chlorophyll, protein, and dry weight (Supplementary Fig. 2b). Please also note that we never refer to cell volume/ population biomass in the manuscript. Thus we think that cell number is sufficient for cell quantification and will not affect subsequent data interpretation.

In the revised manuscript, we have added a caveat stating that cell counting does not take into account that cell size may vary across experimental conditions in this study (see lines 623-624 for detailed changes).

4. L31, "Further sequence alignment analysis of the ion-binding site of the ATP synthase": Since this is a description of the ion selectivity of the c subunit, the term ion-binding site is inappropriate.

Thanks for your kind comment. After careful consideration and discussion, we have decided to remove this sentence from the abstract and to remove the whole section of "Sodium energetics in the cyanobacterial phyla" including corresponding figures (Fig. 7a and Supplementary Fig. 9 of the originally submitted files) and contexts, as the evidence presented seems to be speculative and confusing. Changes can be found in lines 30 and 519.